# SGHormerVQ: Bridging Graph Transformers and Spiking Neural Networks via Spiking Vector Quantization

## Abstract

Graph Transformers (GTs), which simultaneously integrate message passing and self-attention mechanisms, have achieved promising empirical results in some graph prediction tasks. Although these approaches show the potential of Transformers in capturing long-range graph topology information, issues concerning the quadratic complexity and high computing energy consumption severely impair the scalability of GTs on large-scale graphs. Recently, as brain-inspired neural networks, Spiking Neural Networks (SNNs) provide an energy-saving deep learning option with lower computational and storage overhead via their unique spike-based event-driven biological neurons. Inspired by these characteristics, we propose SGHormerVQ, which bridges efficient Graph Transformers and spiking neural networks via spiking vector quantization. Spiking vector quantization generates implied codebooks with smaller sizes and higher codebook usage to assist self-attention blocks in performing efficient global information aggregation. SGHormerVQ effectively alleviates the reliance on complex machinery (distance measure, auxiliary loss, etc.) and the *codebook collapse* present in previous vector quantization-based GNNs. In experiments, we compare SGHormerVQ with other state-of-the-art baselines on node classification datasets ranging from small to large. Experimental results show that SGHormerVQ has achieved competitive performances on most datasets while maintaining up to **518×** faster inference speed compared to other GTs. Our code is available at https://anonymous.4open.science/r/SGHormerVQ-0BB0.

## 1 Introduction

Graph Transformers (GTs), as emerging graph representation learning paradigms, are proposed for alleviating inherent drawbacks present in message passing neural networks like over-smoothing, over-squashing and local structure biases Oono & Suzuki (2019)Topping et al. (2021). Benefiting from the multi-head attention (MHA) modules, vanilla Transformers adaptively learn the global dependencies in input sequences without considering their distance Vaswani (2017). It also provides a solution for learning new topology among nodes while performing message aggregation on the graph data. Experiments demonstrate the immense potential of Transformers in handling global dependencies for graph data Rampášek et al. (2022)Bo et al. (2023). However, there is one critical drawback that Transformer with $O(N^2)$ computation complexity is prohibitive for large-scale graphs. Furthermore, the all-pair similarity matrix leads to an increase in degrees of freedom, which often manifests as the full-size Transformers being highly prone to overfitting. Unlike observations in the field of computer vision or natural language process, previous studies show that eliminating redundant components and embracing a lightweight architecture like linear-time attention can significantly enhance the predictive performances of GTs Wu et al. (2022)Wu et al. (2024).

Recently, with the development of neuromorphic computing, Spiking Neural Networks (SNNs) are poised to bridge the efficiency gap between elaborate network architectures and computation consumption. The defining feature of a SNN is its brain-like spiking mechanism which converts real-value signals into single-bit, sparse spiking signals based on its event-driven biological neurons. The single-bit nature enables us to adopt more addition operations rather than expensive multiply-and-accumulate operations on the spiking outputs, while the sparsity means spikes are cheap to

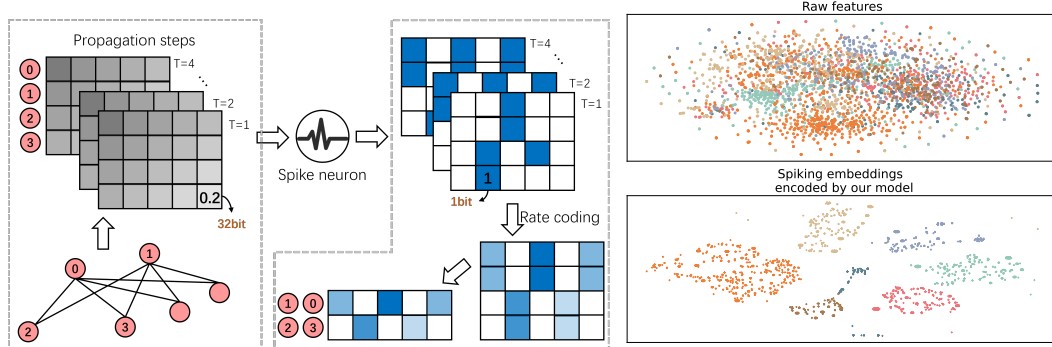

(a) The mechanism of rate-coded spiking neurons.  (b) t-SNE visualization of rate-coded vectors.

Figure 1: (a) On the KarateClub dataset, high-precision neighborhood messages aggregated in each propagation step are converted into sparse, binary spikes. Nodes represented by low-precision rate-coded vectors are implicitly grouped by their neighborhood structure. (b) On the Cora dataset, nodes are colored according to their classes. The t-SNE visualization of learned spiking representations shows that low-precision vectors still maintain promising representational capacity.

store Eshraghian et al. (2023). These delightful characteristics have prompted some studies aiming at constructing lightweight graph representation learning frameworks to explore binary spike-based representations on the graph data Zhu et al. (2022)Li et al. (2023a)Li et al. (2023b). Previous experiments show that the potentiality of SNNs is still underestimated and underappreciated in the domain of graph representation learning. The role of SNNs in modeling the graph structure and dynamics warrants further investigation.

We conduct preliminary experiments to uncover the role of spiking neurons in message propagation. Here, we have three observations: **(i)** For message propagation-based models, the trained embeddings of nodes within the same class tend to exhibit similar distributions. **(ii)** As depicted in Figure1, different messages received from the multiple propagation steps can be converted into the same spike trains via spike neurons. By converting the outputs into firing rates/spike counts, different nodes can be represented by the same low-precision vectors. **(iii)** The precision of generating node representations can be adjusted by configuring spiking neurons. More visualization results are shown in Appendix A. It efficiently encodes continuous high-precision representations into discrete low-precision representations, which sparks our curiosity about an interesting research question.

*For the large-scale graph data, is it possible to eat Graph Transformer cake with a spiking fork?*

Deviating from previous works on efficient GTs, we consider spikes neurons as learnable quantizers, which links the spiking outputs with the concept of vector quantization (VQ) Van Den Oord et al. (2017)Lingle (2023)Mentzer et al. (2023). We propose SGHormerVQ, which bridges spiking neural networks with Graph Transformers via spiking vector quantization. Specifically, we employ spike neurons to capture the message propagation patterns of node neighborhoods, which enables us to represent graph global structure information using a handful of rate-coded vectors. It effectively reduces the reliance of GTs on the full set of node embeddings. Meanwhile, the conversion from real values to spikes implicitly involves the process of learning and generating the codebook in VQ. Different from prior works, the codewords used in practice are governed by learnable parameters in spike neurons. It not only compresses the codebook size, but also addresses the *codebook collapse* which is defined as the under-usage of the codebook. The contributions of this paper are summarized as follows:

- We investigate the role of spiking neurons in the message propagation process. The observations show that nodes can be effectively represented by rate-coded vectors from a discrete subspace of lower dimension, which are transformed from neighborhood messages in the propagation process.
- Based on the observations, we propose Spiking Vector Quantization (SVQ) to replace the pre-defined, fixed codebook with the codebook with variable size. Compared with existing

VQ-based graph representations learning methods, SVQ provides a spike-driven learnable codebook paradigm to alleviate inherent issues in VQ.

- We take the lead in exploring the effectiveness of compressing node representations by spiking neurons in Graph Transformers. We propose the spiked-driven linear-time Graph Transformer, SGHormerVQ. It actively injects the global message propagation patterns in the form of rate-coded vectors to efficiently capture the long-range information.

- We conduct a comprehensive comparison with various state-of-the-art baselines, across graphs of various scales. Extensive experiments show that SGHormerVQ achieves competitive or even superior predictive performances on most datasets. Besides, SGHormerVQ enjoys up to **518x** faster inference speed compared to other GT baselines.

## 2 RELATED WORK

**Spiking Neural Network.**    Inspired by brain-like spiking computational frameworks, Spiking Neural Networks are proposed to address the computing energy consumption challenge. Different from Artificial Neural Networks (ANNs), neurons in SNNs communicate via binary and sparse spikes. It enables SNNs to reduce the storage overhead of intermediate outputs among layers and utilize more accumulation operations instead of multiply-accumulation operations Roy et al. (2019). Some studies motivated by these advantages in energy efficiency have attempted to construct spike-driven neural networks. These networks can be broadly divided into two categories, ANN-to-SNN conversion and direct training framework. For the former, they tend to build a SNN upon a pre-trained ANN. These methods try to minimize information loss during the conversion process by performing scaling/normalizing operations on weights or replacing the activation layers with spike neurons Diehl et al. (2015)Cao et al. (2015)Hao et al. (2023). For the latter, studies try to directly train spike-driven neural networks by introducing surrogate gradient. This approach effectively reduces the strong dependence of SNNs on the number of time steps Fang et al. (2021a)Zheng et al. (2021).

**Graph Transformers.**    Although graph neural networks (GNNs) have become dominant paradigm cross various graph tasks Kipf & Welling (2016)Veličković et al. (2017), the message passing mechanism as the foundations of GNNs has some well-known drawbacks such as over-smoothing, over-squashing and the neglect of long-range information Li et al. (2018)Alon & Yahav (2020). Graph Transformers (GTs), which can differentially aggregate messages over all nodes to alleviate local structure bias, have been developed to overcome above issues. Specifically, methods tend to inject the graph topology information into Transformer variants by introducing auxiliary GNNs or generating positional/structural embeddings from a graph. For some early studies, GTs are proposed to solve small-scale graph-level tasks like molecular property prediction and molecule classification Rampášek et al. (2022)Liao & Smidt (2022). Recently, some methods aimed to enhance GTs' performance on node-level tasks by constructing mini-batch sampling strategies and lightweight attention modules Shirzad et al. (2023)Li et al. (2024).

**VQ-VAE and Follow-up.**    To overcome the issue of posterior collapse, Van Den Oord et al. (2017) develop a discrete latent variational autoencoder (VAE) model called VQ-VAE. Input images are mapped to the embedding space through an encoder. The embeddings are replaced with the nearest pre-defined codebook entries through measuring distances between embeddings and entries. The replaced outputs are then fed into the decoder. It's worth noting that VQ-VAE utilizes a commitment loss and the straight-through estimator to update the codebook and encoder-decoder modules. In addition, Kolesnikov et al. (2022) adopted a codebook splitting algorithm to improve codebook usage. Mentzer et al. (2023) implicitly constructs a finite scalar codebook by quantizing elements of intermediate embeddings to integers. In a nutshell, experiments have shown that VQ-VAE and its variants provide a simpler representation generation schema and an energy-efficient inference framework compared with their quantization-free counterparts.

## 3 PRELIMINARIES

**Spiking Neural Network.**    Although electrophysiological measurements can be accurately calculated by complex conductance-based neurons, the complexity limits their widespread deployment in

deep neural networks. A simplified computational unit that retains biological characteristics, known as the Integrate-and-Fire neuron, has been proposed Salinas & Sejnowski (2002). IF neurons have three basic characteristics: **Integrate**, **Fire** and **Reset**. Firstly, the neuron integrates synaptic inputs from other neurons or external current $I$ to charge its cell membrane. Secondly, when the membrane potential reaches a pre-defined threshold value $V_{th}$, the neuron fires a spike $S$. Thirdly, the membrane potential of neuron will be reset to $V_{reset}$ after firing. The neuronal dynamics can be formulated as follows:

$$\text{Integrate:} \quad V^t = \Psi(V^{t-1}, I^t) = V^{t-1} + I^t, \tag{1}$$

$$\text{Fire:} \quad S^t = \Theta(V^t - V_{th}) = \begin{cases} 1, & V^t - V_{th} \geq 0 \\ 0, & otherwise \end{cases}, \tag{2}$$

$$\text{Reset:} \quad V^t = V^t(1 - S^t) + V_{reset}S^t, \tag{3}$$

where $V^t$ and $I^t$ denote the membrane potential and input current at time step $t$, respectively. $\Theta(\cdot)$ denotes the fire function, and the Heaviside step function is selected as the fire function in this paper. $\Psi(\cdot)$ is the membrane potential update function. Besides, there are two common variants of IF model, LIF and PLIF Gerstner et al. (2014)Fang et al. (2021b). The update function of these neurons can be formalized as follows:

$$\text{LIF:} \quad V^t = V^{t-1} + \frac{1}{\tau}(I^t - (V^{t-1} - V_{reset})), \tag{4}$$

$$\text{PLIF:} \quad V^t = V^{t-1} + \frac{1}{1 + exp(-a)}(I^t - (V^{t-1} - V_{reset})), \tag{5}$$

where $\tau$ is the membrane time constant and $a$ is a trainable parameter, both of which are used to regulate how fast the membrane potential decays. In this paper, we adopt surrogate gradients during error backpropagation to address the issue of zeros gradients caused by non-differentiable functions Neftci et al. (2019). The surrogate gradient method can be defined as $\Theta'(x) \triangleq \theta'(\alpha x)$, where $\alpha$ represents a smooth factor and $\theta(\cdot)$ represents a surrogate function Neftci et al. (2019).

**Graph Neural Network.** We represent a graph as $\mathcal{G} = (\mathcal{V}, \mathcal{E})$, where $\mathcal{V}$ is a set of nodes and $\mathcal{E}$ is a set of edges among these nodes, $\mathbf{A} \in \mathbb{R}^{N \times N}$ is the adjacency matrix of the graph. Let $N$ denotes the number of nodes. We define the $d$-dimension nodes' attribute as $\mathbf{X} \in \mathbb{R}^{N \times d}$, which is known as the node feature matrix. For a given node $u \in \mathcal{V}$, GNN aggregates messages from its immediate neighborhood $N(u)$ and updates the node embedding $h_u$. This message-passing process can be formulated as follows:

$$h_u^l = \text{UPDATE}(h_u^{l-1}, \text{AGGREGATE}(h_v^l, \forall v \in \mathcal{N}(u))), \tag{6}$$

where $h_u^l$ denotes the updated embedding of node $u$ in the $l$-th layer, $h_u^{l-1}$ is the embedding from the previous layer. UPDATE and AGGREGATE can be arbitrary differentiable functions.

**Self-Attention.** As the most prominent component in the Transformer, the self-attention mechanism can be seen as mapping a query vector to a set of key-value vector pairs and calculating a weighted sum of value vectors as outputs. let nodes' attribute $\mathbf{X} \in \mathbb{R}^{N \times d}$ be the input to a self-attention layer. The attention function is defined as follows:

$$Attn(\mathbf{X}) = softmax(\frac{\mathbf{Q}\mathbf{K}^T}{\sqrt{d'}})\mathbf{V}, \tag{7}$$

$$\mathbf{Q} = \mathbf{X}\mathbf{W_q}, \mathbf{K} = \mathbf{X}\mathbf{W_k}, \mathbf{V} = \mathbf{X}\mathbf{W_v}, \tag{8}$$

where Query, Key and Value are calculated by learnable projection matrices $W_q, W_k, W_v \in \mathbb{R}^{d \times d'}$. For the node $u$, the attention function can be written in a message-passing form as:

$$Attn(x_u) = \sum_i^N \frac{exp(q_u^T k_i)}{\sum_j^N exp(q_u^T k_j)} v_i = \sum_i^N \frac{exp((\mathbf{W_q}x_u)^T(\mathbf{W_k}x_i))}{\sum_j^N exp((\mathbf{W_q}x_u)^T(\mathbf{W_k}x_j))}(\mathbf{W_v}x_i), \tag{9}$$

where we omit the scalar factor for brevity.

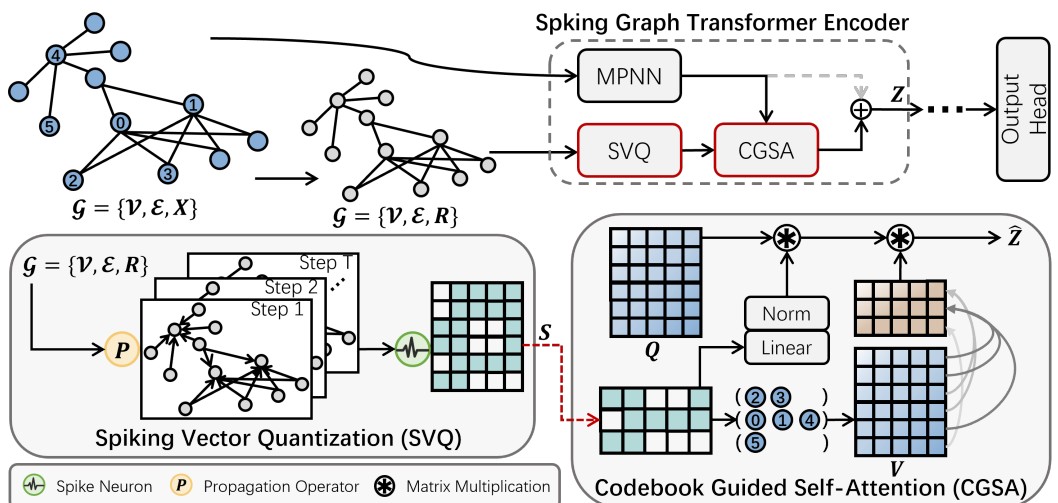

Figure 2: The overview of SGHormerVQ. Intuitively, in the spiking vector quantization block, nodes represented in the form of rate-coded vectors are implicitly grouped by their neighborhood structures. The rate-coded vector can be considered as the codeword corresponding to the node. In the self-attention block, based on the above codewords, the vanilla attention between nodes has been transformed into a linear-time attention from nodes to grouped node sets.

## 4 PRESENT WORK: SGHORMERVQ

In this section, we comprehensively detail our approach referred to as SGHormerVQ. As depicted in Figure 2, SGHormerVQ feeds the graph topology information into Spiking Vector Quantization (SVQ) module to map node embeddings into the rate-coded vectors. The outputs will guide the aggregation process in self-attention. Besides, auxiliary message passing neural networks as position encoders provide node embeddings containing local positional information to the attention module. In what follows, we first highlight the implementation of SVQ (Section 4.1). Then, we detail how the learnable codebook is introduced into self-attention to capture long-range information in the graph (Section 4.2). Finally, we review the entire architecture of SGHormerVQ one by one (Section 4.3).

### 4.1 SPIKING VECTOR QUANTIZATION

As aforementioned above, neighborhood messages of different nodes can be encoded into same rate-coded vectors, which provides compact node representations reflecting the neighborhood structural information. To this end, **(i)** we sample a $D$-dimension random feature matrix $\mathbf{R} \in \mathbb{R}^{N \times D}$ from a uniform distribution. And we define a propagation operator $\mathbf{P}$. **(ii)** Our goal is to collect messages $\mathbf{M} \in \mathbb{R}^{N \times D}$ during iterative propagation process and quantize a sequence of changes $\mathcal{M} = \{\mathbf{M}^0, \mathbf{M}^1, ..., \mathbf{M}^T\}$ into a finite set of codewords. **(iii)** For spiking neurons based on rate coding mechanism, they convert inputs into spike counts $\mathbf{S} = \{s_i\}_i^N \in \mathbb{R}^{N \times D}$. The rate-coded vector $s \in \mathbb{R}^D$ can be seen as a codeword $s \in \tilde{\mathbf{C}}$, where $\tilde{\mathbf{C}}$ denotes an implicit codebook. The implied codebook size is determined by both the number of propagation steps $T$ and the random feature dimension $D$. Considering the case where the total number of spikes is zero, the implied codebook size is given by the product of all channels, $|\tilde{\mathbf{C}}| = (T + 1)^D$. The above process is defined as:

$$\hat{\mathbf{M}}^0 = \mathbf{R}, \quad \hat{\mathbf{M}}^t = \mathbf{P}^t \mathbf{R}, \quad \hat{\mathbf{M}}^t \in \mathbb{R}^{N \times D} \tag{10}$$

$$\mathbf{M}^t = \text{Norm}(\hat{\mathbf{M}}^t), \quad \mathbf{M}^t \in \mathbb{R}^{N \times D} \tag{11}$$

$$\mathbf{S} = \sum_t^T \Theta(\Psi(V^{t-1}, \mathbf{M}^t) - V_{th}), \quad \mathbf{S}^t \in \mathbb{R}^{N \times D} \tag{12}$$

where $\Theta(\cdot)$ and $\Phi(\cdot)$ is membrane potential update and fire function. $Norm(\cdot)$ aims at normalizing output messages to the range of threshold membrane potential $V_{th}$. In the implementation, we adopt

simple $\mathbf{l}_2$ normalization, which can be replaced with some advanced normalization variants from previous works Xu et al. (2021). Besides, we follow a similar approach as in previous work Eliasof et al. (2023), using the graph Laplacian $\mathbf{P} = \mathbf{I} - \mathbf{D}^{-1/2}\mathbf{A}\mathbf{D}^{-1/2}$ or adjacency matrix with self-loops $\mathbf{P} = \mathbf{A} + \mathbf{I}$ as the propagation operator.

## 4.2 CODEBOOK GUIDED SELF-ATTENTION

On the basis of the spike vectors, we propose a codebook guided self-attention (CGSA) with linear complexity to capture long-range signals based on the neighborhood structure similarity. Technically, we follow Kong et al. (2023)Lingle (2023) to utilize a matrix $\hat{\mathbf{K}}$ reconstructed from codebook replace the original matrix $\mathbf{K}$. Specifically, we can dynamically generate a codebook $\mathbf{C} \subseteq \hat{\mathbf{C}}$ by removing duplicate vectors in $\mathbf{S}$. The attention function is defined via:

$$\mathbf{S} = \mathbf{UC}, \qquad\qquad \mathbf{U} \in \mathbb{R}^{N \times B}, \mathbf{C} \in \mathbb{R}^{B \times D} \tag{13}$$

$$\mathbf{G} = \text{Norm}(\text{Linear}(\mathbf{C})), \qquad\qquad \mathbf{G} \in \mathbb{R}^{B \times d'} \tag{14}$$

$$\hat{\mathbf{K}} = \mathbf{UG}, \qquad\qquad \mathbf{K} \in \mathbb{R}^{N \times d'} \tag{15}$$

$$\hat{\mathbf{Z}} = \text{softmax}(\mathbf{Q}\hat{\mathbf{K}}^T)\mathbf{V}, \qquad\qquad \hat{\mathbf{Z}} \in \mathbb{R}^{N \times d'} \tag{16}$$

where $d'$ denotes the dimension of intermediate embeddings, $\mathbf{U}$ is a one-hot matrix, and $|\mathbf{C}| = B$. Different from existing methods, which materialize $\hat{\mathbf{K}}$ using the entire explicit codebook, the dynamically generated codebook in our attention module is much smaller than the implied codebook, $B \ll |\tilde{\mathbf{C}}|$. The codebook calculation is conducted on the integer matrix $\mathbf{S}$, which doesn't bring much computational overhead. Derived from Lingle (2023), the attention weights in eq 16 can be further factored:

$$\hat{\mathbf{Z}} = \text{softmax}(\mathbf{Q}\hat{\mathbf{K}}^T)\mathbf{V} \tag{17}$$

$$= \text{softmax}(\mathbf{Q}(\mathbf{UG})^T)\mathbf{V} \tag{18}$$

$$= \text{Diag}^{-1}(\exp(\mathbf{Q}\mathbf{C}^T)\mathbf{U}^T\mathbf{1})\exp(\mathbf{Q}\mathbf{C}^T)\mathbf{U}^T\mathbf{V} \tag{19}$$

where $\mathbf{1} \in \mathbb{R}^N$. $\mathbf{U}^T\mathbf{1} = \{n_b\}_b^B \in \mathbb{Z}^+$ denotes the number of node embeddings in $\mathbf{C}$ mapped to the same codewords, which can be regarded as a normalization term. The complexity of CGSA is $O(NBd_v)$, where $B \ll N$. It can be considered that computational overhead of CGSA grows linearly with the number of nodes. To avoid generating an excessively large codebook in the initial phase of learning, we perform a truncation strategy. We rank $n_b$ from high to low and select the top $B_{max}$ to generate a truncated codebook, which ensures the efficiency of training our model.

## 4.3 OVERALL FRAMEWORK

As shown in Figure 2, the overview of SGHormerVQ includes four modules: SVQ, auxiliary MPNN, CGSA and a classification head (CH). In SGVQ, we construct random features and spike neurons for each layer. By defining a shared propagation operator, messages among nodes are collected and transformed into node spiking embeddings $\mathbf{S}$. Then an auxiliary MPNN as encoders generates node embeddings with local positional encodings. In the CGSA, the spiking outputs $\mathbf{S}$ and the node embeddings $\mathbf{H}$ are fed into a linear-time self-attention. Different from the vanilla Transformer, we explicitly inject graph inductive bias by coding global structural information into spikes. These four parts can be written as follows:

$$\mathbf{S}^l = \text{SVQ}^l(\mathbf{A}), \qquad\qquad \mathbf{S}^l \in \mathbb{R}^{N \times D} \tag{20}$$

$$\mathbf{H}^l = \text{MPNN}^l(\mathbf{Z}^{l-1}, \mathbf{A}), \qquad\qquad \mathbf{H}^l \in \mathbb{R}^{N \times d'} \tag{21}$$

$$\hat{\mathbf{Z}}^l = \text{CGSA}^l(\mathbf{S}^l, \mathbf{H}^l), \qquad\qquad \hat{\mathbf{Z}}^l \in \mathbb{R}^{N \times d'} \tag{22}$$

$$\mathbf{Z}^l = \text{Linear}(\hat{\mathbf{Z}}^l) + \mathbf{H}^l, \qquad\qquad \mathbf{Z}^l \in \mathbb{R}^{N \times d'} \tag{23}$$

$$\mathbf{Y} = \text{CH}(\mathbf{Z}^L), \tag{24}$$

where $L$ is the number of layers. We choose a simple fully connected layer as the classification head. It has been observed that in vanilla Transformers, projection blocks consisting of Multilayer Perceptrons (MLPs) and normalization layers exacerbate the overfitting problem on large-scale graphs. Therefore, we discard redundant projection layers and retain only the self-attention module and the skip-connection structure He et al. (2016).

Table 1: Classification accuracy(%) on seven datasets. Highlighted are the top first, **second** results.

| Models | Cora | CiteSeer | PubMed | Co-CS | Co-Physics | Arxiv | Products |
|---|---|---|---|---|---|---|---|
| #nodes | 2,708 | 3,327 | 19,717 | 18,333 | 34,493 | 169,343 | 2,449,029 |
| #edges | 10,556 | 9,104 | 88,648 | 163,788 | 495,924 | 1,166,243 | 61,859,140 |
| GCN | $81.6_{\pm 0.4}$ | $71.6_{\pm 0.4}$ | $78.8_{\pm 0.6}$ | $92.5_{\pm 0.4}$ | $95.7_{\pm 0.5}$ | $70.4_{\pm 0.3}$ | $75.7_{\pm 0.1}$ |
| GAT | $83.0_{\pm 0.7}$ | $72.1_{\pm 1.1}$ | $79.0_{\pm 0.4}$ | $92.3_{\pm 0.2}$ | $95.4_{\pm 0.3}$ | $70.6_{\pm 0.3}$ | OOM |
| SGC | $80.1_{\pm 0.2}$ | $71.9_{\pm 0.1}$ | $78.7_{\pm 0.1}$ | $90.3_{\pm 0.9}$ | $93.2_{\pm 0.5}$ | $68.7_{\pm 0.1}$ | $74.2_{\pm 0.1}$ |
| VQGraph | $81.1_{\pm 1.2}$ | $\mathbf{74.5}_{\pm 1.9}$ | $77.1_{\pm 3.0}$ | $93.3_{\pm 0.1}$ | $95.0_{\pm 0.1}$ | $\mathbf{72.4}_{\pm 0.2}$ | $\mathbf{78.3}_{\pm 0.1}$ |
| SpikingGCN | $79.1_{\pm 0.5}$ | $62.9_{\pm 0.1}$ | $78.6_{\pm 0.4}$ | $92.6_{\pm 0.3}$ | $94.3_{\pm 0.1}$ | $55.8_{\pm 0.7}$ | OOM |
| SpikeNet | $78.4_{\pm 0.7}$ | $64.3_{\pm 0.8}$ | $79.1_{\pm 0.5}$ | $93.0_{\pm 0.1}$ | $\mathbf{95.8}_{\pm 0.7}$ | $66.8_{\pm 0.1}$ | $74.3_{\pm 0.4}$ |
| SpikeGCL | $79.8_{\pm 0.7}$ | $64.9_{\pm 0.2}$ | $79.4_{\pm 0.8}$ | $92.8_{\pm 0.1}$ | $95.2_{\pm 0.6}$ | $70.9_{\pm 0.1}$ | OOM |
| SpikeGraphormer | $82.0_{\pm 0.7}$ | $70.5_{\pm 0.6}$ | $71.1_{\pm 0.4}$ | $92.1_{\pm 0.8}$ | $95.7_{\pm 0.3}$ | $70.2_{\pm 0.9}$ | OOM |
| NAGphormer | $79.9_{\pm 0.1}$ | $68.8_{\pm 0.2}$ | $\mathbf{80.3}_{\pm 0.9}$ | $93.1_{\pm 0.5}$ | $95.7_{\pm 0.7}$ | $70.4_{\pm 0.3}$ | $73.3_{\pm 0.7}$ |
| GOAT | $73.3_{\pm 0.3}$ | $68.4_{\pm 0.7}$ | $78.1_{\pm 0.5}$ | $\textcolor{blue}{93.5}_{\pm 0.6}$ | $95.4_{\pm 0.2}$ | $\mathbf{72.4}_{\pm 0.4}$ | $\textcolor{blue}{82.0}_{\pm 0.4}$ |
| NodeFormer | $82.2_{\pm 0.9}$ | $72.5_{\pm 1.1}$ | $79.9_{\pm 1.0}$ | $92.9_{\pm 0.1}$ | $95.4_{\pm 0.1}$ | $59.9_{\pm 0.4}$ | OOM |
| SGFormer | $\mathbf{84.5}_{\pm 0.8}$ | $72.6_{\pm 0.2}$ | $\mathbf{80.3}_{\pm 0.6}$ | $91.8_{\pm 0.2}$ | $95.9_{\pm 0.8}$ | $\textcolor{blue}{72.6}_{\pm 0.1}$ | $72.6_{\pm 1.2}$ |
| SGHormerVQ | $\textcolor{blue}{84.7}_{\pm 0.8}$ | $\textcolor{blue}{74.0}_{\pm 0.5}$ | $\textcolor{blue}{80.6}_{\pm 0.4}$ | $\mathbf{93.4}_{\pm 0.4}$ | $\textcolor{blue}{96.2}_{\pm 0.0}$ | $72.0_{\pm 0.1}$ | $74.8_{\pm 0.4}$ |

## 5 EXPERIMENTS

### 5.1 COMPARISON WITH EXISTING MODELS

In this section, we conduct the experimental evaluation to show the effectiveness of SGHormerVQ on node classification datasets. All experiments are conducted using the same dataset splits presented in prior studies. The Hyperparameters search strategy is adopted on both SGHormerVQ and other baselines to get the optimal combinations of parameters. We perform all models on each dataset 5 times with different random seeds to report the mean and standard deviation. All above experiments are conducted on a single NVIDIA RTX 4090 GPU. The subsequent experiments follow the same settings if not explicitly stated otherwise.

**Datasets.** We evaluate SGHormerVQ on seven datasets including three citation networks Sen et al. (2008) (Cora, CiteSeer, PubMed), two co-author networks Shchur et al. (2018) (Coauthor-CS and Coauthor-Physics ) and two large-scale graphs (ogbn-arxiv and ogbn-products) from the Open Graph Benchmark (OGB) Hu et al. (2020). For citation networks, the data splits adhere to the semi-supervised settings. For co-author networks, we randomly split nodes with train/valid/test ratio as 10%/10%/80%. For datasets from the OGB, we adopt their own standard splits.

**Baselines.** To comprehensively evaluate the performance of SGHormerVQ, a head-to-head comparison is conducted with state-of-the-art GNNs and GTs, based on their architectures. As shown in Table 2, components in baselines are fall into three categories: spike-based methods (SpikingGCN Zhu et al. (2022), SpikeNet Li et al. (2023a), SpikeGCL Li et al. (2023b), SpikeGraphormer Sun et al. (2024)), Graph Transformer framework (NAGphormer Chen et al. (2022), NodeFormer Wu et al. (2022), SGFormer Wu et al. (2024), GOAT Kong et al. (2023)), vector quantization-based methods (VQ-

Table 2: Comparison of Graph Transformers and Graph Neural Networks w.r.t. required components (**SP**: spike-based, **GT**: Graph Transformer framework, **VQ**: vector quantization-based).

| Model | Components | | |
|---|---|---|---|
| | SP | GT | VQ |
| SpikingGCNZhu et al. (2022) | ✓ | - | - |
| SpikeNetLi et al. (2023a) | ✓ | - | - |
| SpikeGCLLi et al. (2023b) | ✓ | - | - |
| SpikeGraphormerSun et al. (2024) | ✓ | ✓ | - |
| NAGphormerChen et al. (2022) | - | ✓ | - |
| NodeFormerWu et al. (2022) | - | ✓ | - |
| SGFormerWu et al. (2024) | - | ✓ | - |
| GOATKong et al. (2023) | - | ✓ | ✓ |
| VQGraphYang et al. (2024) | - | - | ✓ |
| **SGHormerVQ** | ✓ | ✓ | ✓ |

Graph Yang et al. (2024)). Besides, three classic graph neural networks (GCN Kipf & Welling (2016), GAT Veličković et al. (2017), SGC Wu et al. (2019)) are also included in the comparison.

**Overall performance.** The experimental results are demonstrated in Table 1. As shown in the table, our methods achieve competitive performance on all datasets, which is a significant advancement considering the information loss caused by low-precision spiking embeddings. **SGHormerVQ outperforms other spike-based baselines across all datasets, which achieves an average improvement of 1.4%.** Furthermore, SGHormerVQ achieves predictive performance on par or even better than high-precision GT methods. SGHormerVQ achieves the best mean Accuracy on Cora, PubMed and Physics. Meanwhile, we also notice that SGHormerVQ falls short of the current sota baseline on the ogbn-products datasets. Here, we present our analysis that the average degree of nodes in ogbn-products is around 50, while it ranges from 3 to 14 in other datasets. For those graphs with abundant neighborhood messages, the spiking encoding and corresponding vector quantization schema exacerbate the information loss together. We leave reducing the information loss in graphs with abundant connectivity for future work. Overall, the results indicate that integrating spiking vector quantization with codebook guided self-attention enables SGHormerVQ to capture long-range node information. It effectively alleviates the impact of information loss caused by the conversion from real values to spikes.

## 5.2 INFERENCE TIME ELAPSE AND ACCURACY

To examine the efficiency of SGHormerVQ, we explore the trade-off between the inference time elapse and prediction performance among GTs. As depicted in Figure 3, SGHormerVQ has achieved the highest accuracy (96.2%) and the fastest inference speed (21ms) among GT baselines on the Physics dataset. Furthermore, in Appendix B, we provide comprehensive energy efficiency analyses between SGHormerVQ and the other GT baselines based on three metrics: the inference latency, maximum memory usage and theoretical energy consumption. The results show that **SGHormerVQ achieves the lowest inference latency across datasets with various scales. Compared to another VQ-based GT, SGHormerVQ with better performance infers faster than GOAT by 518x on the Physics dataset.** The pregenerated codebook in SVQ and linear-time attention modules bring a significant improvement in inference speed. Many previous VQ-based methods tend to replace node representations one by one with learned codewords during the inference phase. In SVQ, trained spiking neurons directly convert input features into codewords, which means the codewords corresponding to nodes can be pre-calculated before the inference phase. As mentioned in the previous section, the complexity of CGSA is

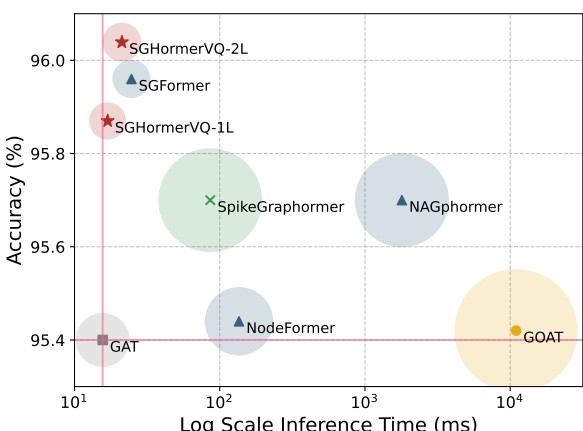

Figure 3: Accuracy versus Inference Time. The size of the circle indicates the maximum memory usage during model training.

$O(NBd_v)$. SGHormerVQ reconstructs a more compact codebook from the outputs of spiking neurons, rather than setting a fixed codebook. The inference time elapse of SGHormerVQ is similar to that of the representative linear-time, SGFormer. **Benefiting from the fusion of SVQ and CGSA, SGHormerVQ outperforms another spike-based GT across all datasets with acceptable theoretical energy consumption.** In Figure 3, a 1-layer SGHormerVQ still rapidly captures local and global graph information while bringing inference speed closer to that of standard GNNs like GAT.

## 5.3 CHARACTERISTICS OF SPIKING VECTOR QUANTIZATION

For elaborately analyzing the spiking vector quantization, we conduct a series of experiments on the SGHormerVQ. In detail, we explore the *codebook collapse* problem in VQ-based graph models.

Table 3: Codebook analysis on Cora and CS datasets. For each, we compare SGHormerVQ with the other two VQ-based graph models, VQGraph and GOAT. Three metrics are tracked, the number of used codewords (CW), codebook usage (Usage) and accuracy (ACC).

| Models | Codebook Size | Cora | | | CS | | |
|---|---|---|---|---|---|---|---|
| | | CW | Usage(%) | ACC(%) | CW | Usage(%) | ACC(%) |
| VQGraph | $2^9$ | 159 | 31.0 | $80.5_{\pm0.2}$ | 84 | 16.4 | $92.7_{\pm0.1}$ |
| | $2^{10}$ | 172 | 16.7 | $80.9_{\pm1.0}$ | 90 | 8.7 | $68.3_{\pm0.6}$ |
| | $2^{11}$ | 186 | 9.0 | $80.4_{\pm1.3}$ | 94 | 4.5 | $71.7_{\pm0.3}$ |
| | $2^{12}$ | 206 | 5.0 | $\mathbf{81.4}_{\pm1.1}$ | 95 | 2.3 | $72.2_{\pm0.3}$ |
| | $2^{13}$ | 284 | 3.4 | $80.9_{\pm0.2}$ | 98 | 1.2 | $68.6_{\pm0.2}$ |
| GOAT | $2^9$ | 89 | 17.3 | $66.8_{\pm0.2}$ | 49 | 9.6 | $90.6_{\pm0.8}$ |
| | $2^{10}$ | 98 | 9.6 | $68.3_{\pm0.9}$ | 100 | 9.8 | $92.2_{\pm0.3}$ |
| | $2^{11}$ | 102 | 5.0 | $71.7_{\pm0.1}$ | 122 | 5.9 | $92.7_{\pm0.7}$ |
| | $2^{12}$ | 100 | 2.4 | $72.2_{\pm0.4}$ | 154 | 3.7 | $92.4_{\pm1.0}$ |
| | $2^{13}$ | 103 | 1.2 | $68.6_{\pm0.1}$ | 162 | 1.9 | $\mathbf{93.4}_{\pm0.3}$ |
| SGHormerVQ | T=4,D=4 | 46 | 100.0 | $80.5_{\pm0.4}$ | 106 | 100.0 | $92.6_{\pm0.8}$ |
| | T=6,D=4 | 87 | 100.0 | $80.6_{\pm0.9}$ | 274 | 100.0 | $92.8_{\pm0.8}$ |
| | T=4,D=6 | 122 | 100.0 | $\mathbf{81.8}_{\pm1.5}$ | 238 | 100.0 | $93.0_{\pm0.1}$ |
| | T=4,D=8 | 104 | 100.0 | $\mathbf{81.4}_{\pm0.5}$ | 328 | 100.0 | $\mathbf{93.1}_{\pm0.3}$ |

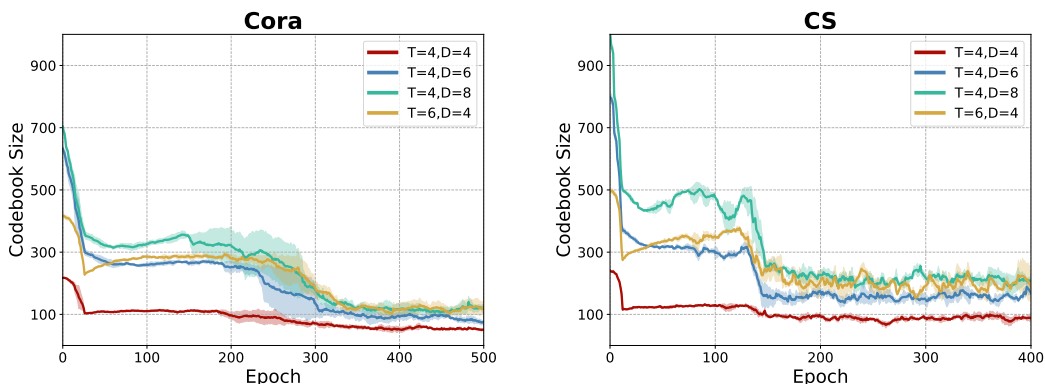

Figure 4: The number of used codewords in the training step.

We record the number of used codewords from the implied codebook during the training process of our model, and investigate the codebook usage among VQ-based graph methods to explore the following questions: (ii) How does the implicit codebook influence our model? (i) Is the spiking vector quantization a more efficient VQ alternative?

**Factors affecting the codebook size.** As aforementioned above, the number of propagation steps $T$ and the random feature dimension $D$ determine the implied codebook size. In Figure 4, we construct 4 combinations of these two hyperparameters ($T = 4/D = 4$, $T = 6/D = 4$, $T = 4/D = 6$, $T = 4/D = 8$), which aims at matching pre-defined codebook size ($2^8$, $2^{11}$, $2^{12}$, $2^{13}$). We observe that injecting the graph inductive bias as a kind of prior knowledges to quantizers does constrain the size of the codebook, ensuring convergence during the learning process. However, increasing the size of the implicit codebook does not effectively improve the codebook usage in training process. Table 3 shows that an excessively large implied codebook impairs performances of SGHormerVQ on the small-scale dataset. $T$ significantly influences the complexity of spike patterns, thereby affecting the number of used codewords.

**Codebook Usage.** In Table 3, we exhibit the codebook usage of multiple VQ-based graph methods, which is defined as the fraction of used codewords. It suggests that those methods using pre-defined codebooks suffer from the serious issue of *codebook collapse*. As the codebook size

increases, this issue becomes more pronounced. For GOAT, the average codebook usages are 7.1% and 6.1% on Cora and CS datasets. The codebook usages in VQGraph are slightly higher, which achieve 13% and 6.6%. Although the number of used codewords does increase, it is an inefficient way that creating an excessively large codebook to improve the performance on large-scale graphs. As a more efficient solution, SHormerVQ constructs an implicit codebook governed by spike neurons, which brings 100% codebook usage. In some cases, the number of used codewords in SGHormerVQ will be slightly larger than vanilla VQ counterparts at the beginning of the training process, we believe this issue can be effectively mitigated by designing appropriate spike neurons.

## 5.4 ABLATION STUDY

In this section, we conduct ablation studies to analyze the differences between different linear-time attention mechanisms and explore the impact of different spike neurons on predictive performances. To this end, we implement two classic linear-time attention modules (Performer Choromanski et al. (2020) and Linformer Wang et al. (2020)), two common spike neurons (IF and LIF), two normalization algorithms (LayerNorm Ba et al. (2016) and STFNorm Xu et al. (2021)) and remove SVQ modules to construct 6 SGHormerVQ variants.

The experimental results are demonstrated in Table 4. Although incorporating extra positional encodings from MPNNs enables Performer and Linformer to handle graph prediction tasks, they struggle to achieve good predictive performance on large-scale graphs like ogbn-arxiv. In SGHormerVQ, the CGSA actively introduces the global structure information during attention score calculation. It suggests that developing graph structure-aware Transformers is a promising direction for scaling GTs on large-scale graphs. The choice of spike neurons will affect the predictive performances of SGHormerVQ. PLIF models, which have learnable membrane time constants and synaptic weights, achieve slightly better performance in most cases. These neurons effectively endow SVQ with better flexibility. In addition, the well-designed normalization algorithm for spiking neurons, STFNorm, outperforms the LayerNorm algorithm across all datasets. For spiking graph neural networks, the distribution of spiking node representations and corresponding normalization algorithms lack further exploration. We leave the designs of specific spike neurons and normalization layers on the graph data for future work.

Table 4: Ablation studies on Pubmed, CS, Physics and ogbn-arxiv datasets. $-x$ means removing the component $x$ from SGHormerVQ. And $+x$ means replacing the original component in SGHormerVQ with $x$.

| Models | Pubmed | CS | Physics | ogbn-arxiv |
|---|---|---|---|---|
| +Performer | $80.2_{\pm 0.2}$ | $93.1_{\pm 0.4}$ | $95.8_{\pm 0.1}$ | $71.2_{\pm 0.1}$ |
| +Linformer | $79.6_{\pm 1.0}$ | $92.6_{\pm 0.5}$ | $95.5_{\pm 0.1}$ | $65.2_{\pm 1.3}$ |
| +IF | $81.6_{\pm 1.2}$ | $92.8_{\pm 0.1}$ | $96.0_{\pm 0.4}$ | $71.0_{\pm 0.5}$ |
| +LIF | $79.6_{\pm 0.7}$ | $92.8_{\pm 0.0}$ | $96.1_{\pm 0.2}$ | $72.1_{\pm 0.2}$ |
| +LayerNorm | $78.9_{\pm 1.3}$ | $90.3_{\pm 0.6}$ | $95.4_{\pm 0.4}$ | $71.2_{\pm 0.2}$ |
| +STFNorm | $82.6_{\pm 0.2}$ | $92.3_{\pm 0.4}$ | $96.5_{\pm 0.5}$ | $72.4_{\pm 0.7}$ |
| -SVQ | $79.8_{\pm 0.4}$ | $93.2_{\pm 0.3}$ | $95.0_{\pm 0.4}$ | $70.7_{\pm 0.7}$ |
| SGHormerVQ | $80.6_{\pm 0.5}$ | $93.4_{\pm 0.1}$ | $96.2_{\pm 0.0}$ | $72.0_{\pm 0.1}$ |

## 6 CONCLUSION

In this study, we propose SGHormerVQ, a linear-time Graph Transformer via spiking vector quantization. Based on the observation that the message propagation patterns of different nodes can be encoded into same rate-coded vectors, SGHormerVQ bridges Graph Transformer with spiking neural networks. It enables SGHormerVQ to achieve less information loss, faster inference speed and better predictive performance. Besides, spike vector quantization, which treats spike neurons as quantizers, provides a spiking perspective to address issues present in current VQ methods. We believe that our work holds great promise from a neuroscientific perspective, and we hope it will inspire further research into more efficient Graph Transformers.

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

## A  VISUALIZATION RESULTS OF SVQ

To better demonstrate our observations, we remove spiking neurons in SVQ and construct a simplified message propagation model on the KarateClub dataset. The initial node features are sampled from a uniform distribution over the interval $(0, 1)$. Setting the number of propagation steps to 2, we visualize the message embeddings from each propagation step in the left plots of Figure 5. **It shows that as the number of propagation steps increases, the neighborhood message embeddings of nodes in the same class become increasingly similar. It implies that we can generate the same representation for different nodes by capturing the dynamics in the propagation process.** Benefiting from the powerful coding mechanism of SNNs for sequential data, we fed

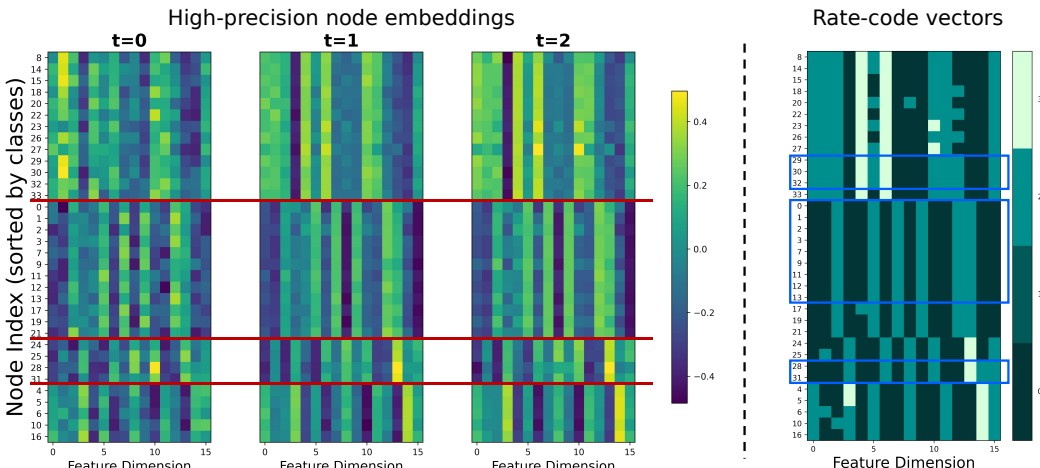

Figure 5: The visualization results between high-precision node embeddings output from each propagation step and the low-precision rate-coded vectors. The feature dimension is set to 16, and nodes are sorted by their categories. The red line is used to differentiate nodes in different categories, and the nodes within the blue box have the same rate-coded vectors. Brighter spots denote higher values.

the above intermediate embeddings into spiking neurons to generate node representations based on spike counts. As shown in the right plot in Figure 5, **different nodes are represented by the same rate-coded vector, which means high-precision node embeddings can be encoded into a finite set of rate-coded vectors from narrower and discrete representation space.**

Furthermore, we perform the SVQ defined in Section 4.1 and the non-spiking counterpart on Cora, Citeseer and Pubmed datasets. In the implementation, the random features will serve as the initial membrane potential of spiking neurons. SVQ updates node representation by alternating propagation and normalization operations, and the symmetrized graph Laplacian and $l_2$ normalization are selected as the propagation operator and normalization function. Visualization results are shown in Figure 6. The high-precision node representations (the leftmost plot in each line) can be projected into the finite set of low-precision rate-coded vectors (the two rightmost plots in each line). Considering iteratively propagated messages as input currents of spiking neurons will generate expressive low-precision node vectors. It explains why some emerging spiking graph neural networks (SGNN) Li et al. (2023a)Yin et al. (2024) achieve better predictive performance compared to earlier approaches Zhu et al. (2022) that rely on repeatedly passing the same training graph data. Additionally, visualization results reveal **the precision of rate-coded vectors is governed by spiking neurons with different configurations.** Higher threshold potentials always correspond to lower fire rates or spike counts, which indirectly drives SGNNs to generate node representation with lower precision.

## B    ENERGY EFFICIENCY ANALYSIS

To verify the efficiency of SGHormerVQ, we conduct energy efficiency analysis on CS, Physics, ogbn-arxiv and ogbn-products datasets based on following metrics: the maximum memory usage, inference latency and theoretical energy consumption. We record the absolute elapsed running time per test epoch for SGHormerVQ and other GT baselines. Notably, following the same settings as previous studies Wu et al. (2024), we use the mini-batch partition for training on the ogbn-products dataset.

The theoretical energy consumption estimation is derived from Yao et al. (2024). For the sake of fairness in comparison, we fix some hyperparameters like the number of layers, the number of heads and the dimension of hidden embeddings for each model. The theoretical energy consumption of GTs during the inference phase is estimated in a straightforward way by counting floating point operations (FLOPs) and synaptic operations (SOPs). As depicted in Figure 3, we can deploy the spiking vector quantization module driven by spiking neurons on the specific neuromorphic hard-

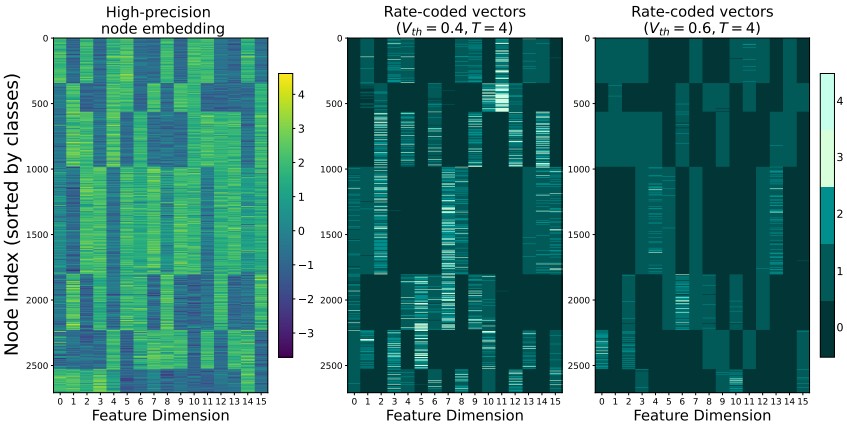

(a) The spike count visualization on Cora.

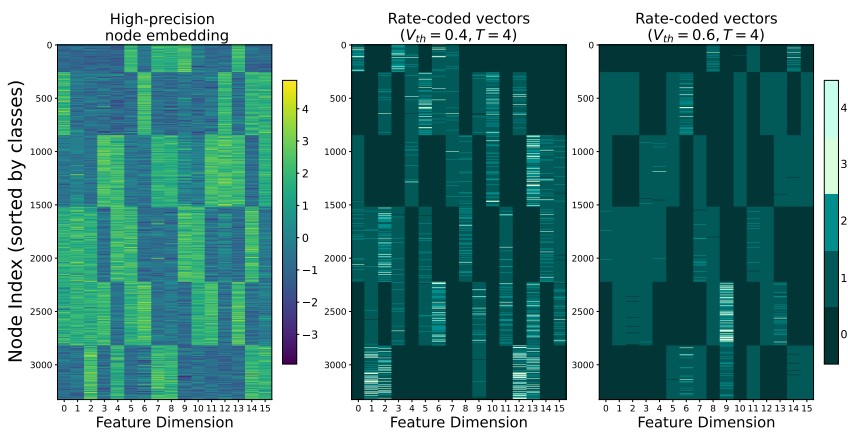

(b) The spike count visualization on Citeseer.

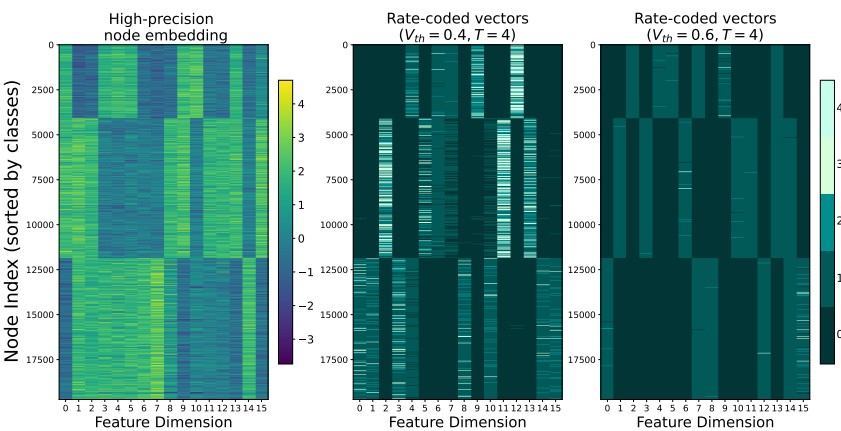

(c) The spike count visualization on Pubmed.

Figure 6: The visualization results of node representations which are decoded in the form of the spike count. The feature dimension is set to 16 and nodes are sorted by their categories. Brighter spots denote higher spike counts.

ware. Therefore, the energy cost of SGHormerVQ can be formulated as follows:

$$E = \sum_{l=1}^{L} (E_{SVQ} + E_{CGSA} + E_{MPNN} + E_{Linear}) + E_{CH} \quad (25)$$

$$= \alpha_s (\sum_{l=1}^{L} \sum_{t=1}^{T} SP_{SVQ}^{l,t}) + \alpha_f (\sum_{l=1}^{L} (FP_{CGSA}^l + FP_{MPNN}^l + FP_{Linear}^l) + FP_{CH}) \quad (26)$$

Table 5: The maximum memory usage (MB), theoretical energy consumption (J) and inference latency (s) of various GT methods.

| Datasets | Metrics | NAGphormer | GOAT | NodeFormer | SGFormer | SpikeGraphormer | SGHormerVQ |
|---|---|---|---|---|---|---|---|
| CS | Latency↓ | 0.70 | 5.02 | 0.05 | 0.01 | 0.03 | 0.01 |
| | Memory↓ | 3400 | 12490 | 2822 | 1662 | 8542 | 1638 |
| | Energy↓ | 0.82 | 1.21 | 0.21 | 0.35 | 0.12 | 0.16 |
| Physics | Latency↓ | 1.79 | 10.98 | 0.14 | 0.02 | 0.08 | 0.02 |
| | Memory↓ | 13628 | 22776 | 7624 | 2944 | 16414 | 3036 |
| | Energy↓ | 1.86 | 2.35 | 0.46 | 0.78 | 0.27 | 0.36 |
| arXiv | Latency↓ | 0.78 | 28.27 | 1.17 | 0.10 | 0.30 | 0.08 |
| | Memory↓ | 10450 | 21146 | 11988 | 6386 | 22654 | 7132 |
| | Energy↓ | 1.12 | 9.92 | 0.63 | 0.57 | 0.08 | 0.18 |
| Products | Latency↓ | 25.74 | 2416.84 | - | 24.34 | - | 20.83 |
| | Memory↓ | 7470 | 21974 | - | 934 | - | 13494 |
| | Energy↓ | 16.06 | 143.80 | - | 8.07 | - | 3.67 |

where $\alpha_f$ and $\alpha_s$, as scale factors for floating point and synaptic operations, are set to 4.5 and 0.9. $FP$ and $SP$ are denoted as floating point operations and synaptic operations of each layer. $SP^{l,t} = r^{l,t} \times FLOP^{l,t}$, where $r^{t,l}$ is the fire rate of spiking neurons in the $l$-th layer at the $t$-th time step. **The results in table 5 show that SGHormerVQ achieves the fastest inference speed across all datasets compared to other baselines.** Notably, we can generate and store codewords corresponding to each node on the neuromorphic hardware. It enables SGHormerVQ to maintain the codebook with relatively low energy consumption. Additionally, as shown in Figure 4, the size of the reconstructed codebook, $B$, will be gradually decreased during the training process. It makes the linear-time Transformer guided by this compressed codebook infer slightly faster than SGFormer, while bringing a slight extra energy cost compared to SpikeGraphormer.

## C    RATE VERSUS TEMPORAL CODING

The rate coding is the foundation of most spiking graph neural networks because this coding mechanism is quite convenient to integrate with an artificial neural network architecture. As mentioned in the above section, it can convert input intensity into a spike count or firing rate Eshraghian et al. (2023). However, the information loss caused by the rate coding can't be overlooked for directly training SNNs. Some emerging studies focus more on another coding strategy based on the precise timing of a spike. For example, GRSNN Xiao et al. (2024) introduces spiking time as supplementary information to encode relations in knowledge graphs. The empirical experiments verify the efficacy of adding synaptic delays to different edges in message propagation. It drives us to explore the spiking vector quantization based on temporal coding. Since GRSNN is designed for link prediction tasks and the properties of edges are plain on existing node classification datasets, we assign random features to nodes except for the embedding of relations. For node classification tasks, the outputs combined edge embeddings containing the temporal delay information with node embeddings will be fed into a mean aggregator to generate the predictive results. The visualization results of intermediate node embeddings from RGSNN and SVQ are demonstrated in Figure 7. GRSNN considering temporal delays in output spikes does provide more expressive rate-coded vectors. However, the process of reconstructing high-precision node embeddings conflicts with SVQ, which aims at mapping different nodes into similar low-precision rate-coded vectors. Other simpler temporal coding algorithms like the time-to-first-spike mechanism may impede learning convergence due to the lack of sufficient spikes.

## D    REVISITING SGHORMERVQ IN THE PERSPECTIVE OF HOMOPHILY

There is a popular notion that message propagation-based methods are more suitable for graphs with high-level homophily Ma et al. (2021). Therefore, in this section, we conduct a quantitative analysis to investigate whether the homophily of graphs is a determining factor on the performance of SGHormerVQ. Specifically, we perform the same graph generation strategy on Cora and Citeseer datasets following the previous study Ma et al. (2021). Figure 8 shows the influ-

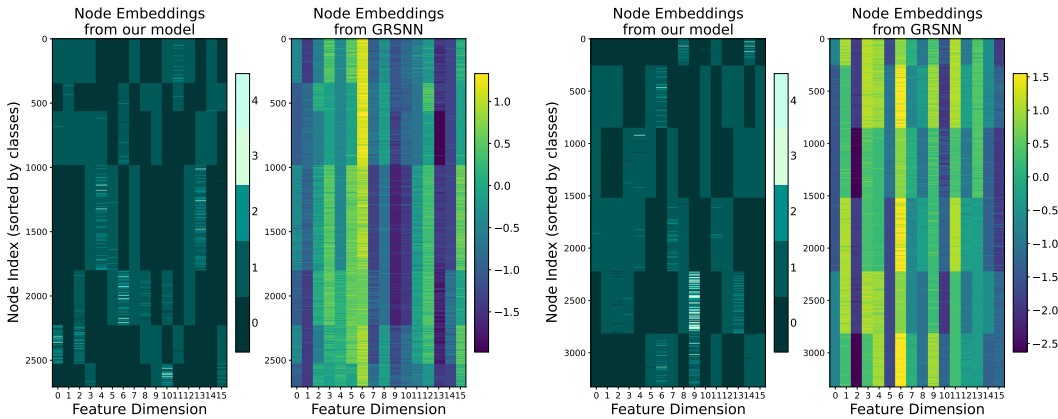

(a) Rate and temporal coding embeddings on Cora.    (b) Rate and temporal coding embeddings on Citeseer.

Figure 7: The visualization results of rate and temporal coding embeddings

ences of different homophily ratios on predictive results, where the homophily ratio is defined as $\frac{|\{(v,w):(v,w)\in\mathcal{E}\cap y_v=y_w\}|}{|\mathcal{E}|}$. We find that as the homophily ratio decreases, the classification performance initially declines but eventually starts to improve. It is consistent with previous observations Ma et al. (2021) that the homophily assumption of message passing-based methods is not accurate. And it implies SGHormerVQ may achieve strong performances on certain heterophilic graphs. Furthermore, we evaluate SGHormerVQ on two heterophilic datasets, Actor and Deezer. The results in Table 6 show that SGHormerVQ has the best classification accuracy on the Actor dataset compared with other baselines. Empirical results highlight the efficacy of SGHormerVQ on both heterophilic and homophilic graphs.

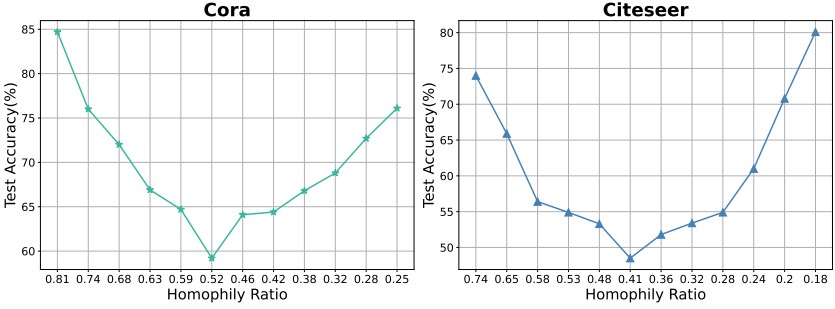

Figure 8: The accuracy of SGHormerVQ on synthetic graphs (Cora and Citeseer) with various homophily ratios.

Table 6: Classification accuracy(%) on two heterophilic datasets (Actor and Deezer).

| Models | Actor | Deezer |
|---|---|---|
| #nodes | 7,600 | 28,281 |
| #edges | 30,019 | 185,504 |
| GCN | $30.1_{\pm 0.2}$ | $62.7_{\pm 0.7}$ |
| GAT | $29.8_{\pm 0.6}$ | $61.7_{\pm 0.8}$ |
| SGC | $27.0_{\pm 0.9}$ | $62.3_{\pm 0.4}$ |
| VQGraph | $\mathbf{38.7}_{\pm 1.6}$ | $65.1_{\pm 0.2}$ |
| SpikingGCN | $26.8_{\pm 0.1}$ | $58.2_{\pm 0.3}$ |
| SpikeNet | $36.2_{\pm 0.9}$ | $65.0_{\pm 0.2}$ |
| SpikeGCL | $30.3_{\pm 0.5}$ | $65.0_{\pm 1.1}$ |
| SpikeGraphormer | $36.0_{\pm 0.5}$ | $65.6_{\pm 0.2}$ |
| NAGphormer | $33.0_{\pm 0.9}$ | $64.4_{\pm 0.6}$ |
| GOAT | $37.5_{\pm 0.7}$ | $65.1_{\pm 0.3}$ |
| NodeFormer | $36.9_{\pm 1.0}$ | $\mathbf{66.4}_{\pm 0.7}$ |
| SGFormer | $37.9_{\pm 1.1}$ | $67.1_{\pm 1.1}$ |
| SGHormerVQ | $\mathbf{39.1}_{\pm 0.2}$ | $65.7_{\pm 0.1}$ |

