# OpenReview forum: "SGHormerVQ: Bridging Graph Transformers and Spiking Neural Networks via Spiking Vector Quantization"
_ICLR.cc/2025/Conference — Submitted to ICLR 2025_

### Official Review · Reviewer_NQbN · 2024-10-30

**Soundness:** 2
**Presentation:** 4
**Contribution:** 2
**Rating:** 6
**Confidence:** 4

**Summary:**

The paper introduces SGHormerVQ, a Graph Transformer model that integrates spiking neural networks through spiking vector quantization (SVQ) to enhance efficiency in graph learning tasks. By using spiking neurons as quantizers, SGHormerVQ achieves faster inference and reduces computational overhead while maintaining competitive performance on node classification benchmarks.

**Strengths:**

The paper is very well-written and looks into an important research question.

**Weaknesses:**

1. I am not sure what is the key novelty of the work - it seems the authors have just used Spiking Vector Quantization (which is itself not a new concept) for spiking graph transformers (also not new)

2. The experimental section is very weak as most of the datasets used for the results are very simple datasets.

3. The paper gives a lot of context and discusses a lot about the prior works. i would recommend to push them to the supplementary section and add more experimental results to show where this model actually works and where it fails- as that is not really clear from the experiments

**Questions:**

1. I am confused by the statement "SGHormerVQ with better performance infers faster thanotherGTs by up to 518x" as the Fig 4 shows that it has higher latency than the standard GAT with slightly higher accuracy? Also, the authors only show this for the simplest dataset (Physics dataset) - I would recommend the authors to also show the latency results for the more complex datasets. Also, how are you getting the latency numbers? Is it just the average for all the inferences in the test set?

2. I would recommend the authors to give a detailed analysis of energy and memory usage would give solid proof of the paper's efficiency claims.The authors could measure the average number of spikes per inference and that would be a good measure for the memory reductions, especially compared to standard Graph Transformers and SNN-based models.

3. I would also recommend the authors further break down the SVQ module to explore the effects of specific components (e.g., varying quantization levels, spike neuron types, or normalization techniques) could highlight why SVQ is particularly effective and reveal which configurations yield optimal performance.

4. I am curious - does the degree of homophily play a role in the SVQ?

---

> ### Author Response · Authors · 2024-11-22
> **Response to Reviewer NQbN**
>
> We thank the reviewer for reading our paper.
>
> > **(W1:)** I am not sure what is the key novelty of the work - it seems the authors have just used Spiking Vector Quantization (which is itself not a new concept) for spiking graph transformers (also not new)
>
> **R1:** Firstly, We would like to emphasize the distinction between our current method and previous approaches that involve the concept of spiking vector quantization [1][2]. In these studies, **constructing or generating the codebook does not involve any strong inductive biases** and **the problem of Codebook Collapse is rarely discussed**. We designed SGHormerVQ to address these issues. The SVQ module actively captures the dynamic during the message propagation to generate a codebook with graph inductive bias. Compared with methods based on pre-defined codebooks, SVQ can be seen as finding the optimal mapping function from nodes to codewords.
>
> Most spiking graph transformers tend to consider spiking neurons as low-power units. The role of spiking neurons in spiking graph neural networks has been insufficiently explored. SGHormerVQ takes it a step further. **In this work, we summarize and compare the strengths and weaknesses of previous SGNN designs (Appendix A).** It suggests that spiking neurons not only are the foundation of low-power models but also capture the dynamics during message propagation. **The architecture we proposed achieves efficiency improvement through spiking vector quantization, while indirectly incorporating graph structural information into the Transformer in the form of codewords to improve the predictive performances.**
>
> > **(W2), (W3):** "The experimental section is very weak as most of the datasets used for the results are very simple datasets.", "The paper gives a lot of context and discusses a lot about the prior works.i would recommend to push them to the supplementary section and add more experimental results to show where this model actually works and where it fails- as that is not really clear from the experiments"
>
> **R2:** Thanks for the suggestions. In experiments, most datasets are consistent with previous studies [3][4]. And the primary goal of our experiments is to verify the effectiveness of the proposed spiking vector quantization and evaluate the energy efficiency of CGSA. **We update a comprehensive energy efficiency analysis in Appendix B and evaluate our methods on two heterophilic datasets (Actor and Deezer) in Appendix D**. Besides, we have updated more explanations and descriptions of the experiments in the related sections.
>
> | Models          | Actor    | Deezer   |
> |-----------------|----------|----------|
> | GCN             | 30.1     | 62.7     |
> | GAT             | 29.8     | 61.7     |
> | SGC             | 27.0     | 62.3     |
> | VQGraph         | 38.7     | 65.1     |
> | SpikingGCN      | 26.8     | 58.2     |
> | SpikeNet        | 36.2     | 65.0     |
> | SpikeGCL        | 30.3     | 65.0     |
> | SpikeGraphormer | 36.0     | 65.6     |
> | NAGphormer      | 33.0     | 64.4     |
> | GOAT            | 37.5     | 65.1     |
> | NodeFormer      | 36.9     | **66.4** |
> | SGFormer        | 37.9     | 67.1     |
> | SGHormerVQ      | **39.1** | 65.7     |

---

> ### Author Response · Authors · 2024-11-22
> **Response to Reviewer NQbN**
>
> > **(Q1):** I am confused by the statement "SGHormerVQ with better performance infers faster than other GTs by up to 518x" as the Fig 4 shows that it has higher latency than the standard GAT with slightly higher accuracy? Also, the authors only show this for the simplest dataset (Physics dataset) - I would recommend the authors to also show the latency results for the more complex datasets. Also, how are you getting the latency numbers? Is it just the average for all the inferences in the test set?
>
> **R3:** Thank you for your helpful advice. Compared to GOAT, SGHormerVQ with better performance infers faster than the VQ-based GT by 518x on the Physics dataset. We apologize for the confusion and have revised this unclear description. In the implementation, **we record the absolute elapsed running time per test epoch as inference latency for SGHormerVQ and other GT baselines. Following the same settings in the previous study [4], we use the mini-batch partition for training on the ogbn-products dataset.** Corresponding experiments about memory usage, inference latency and theoretical energy consumption are provided in Appendix B. Here, we present inference latency (s) results on CS, Physics, arXiv and Products datasets. The results show that the inference latency of SGHormerVQ is comparable to or even better than SGFormer.
>
> | Models          | CS   | Physics | arXiv | Products |
> |-----------------|------|---------|-------|----------|
> | NAGphormer      | 0.7  | 1.79    | 0.78  | 25.74    |
> | GOAT            | 5.02 | 10.98   | 28.27 | 2416.84  |
> | NodeFormer      | 0.05 | 0.14    | 1.17  | -        |
> | SGFormer        | **0.01** | **0.02**    | 0.10  | 24.34    |
> | SpikeGraphormer | 0.03 | 0.08    | 0.30  | -        |
> | SGHormerVQ      | **0.01** | **0.02**    | **0.08**  | **20.83**    |
>
>
> **(Q2):** I would recommend the authors to give a detailed analysis of energy and memory usage would give solid proof of the paper's efficiency claims. The authors could measure the average number of spikes per inference and that would be a good measure for the memory reductions, especially compared to standard Graph Transformers and SNN-based models.
>
> **R4:** Thank you for the suggestions. **We provide a detailed analysis of energy and memory usage in Appendix B. The theoretical energy consumption is calculated by measuring the firing rate [5].** Furthermore, we detail the theoretical energy consumption of SGHormerVQ in the formulation. Energy and memory usage are as follows. It shows that SGHormerVQ with better performances bringing a slight extra energy cost compared to pure SNN-based GT, SpikeGraphormer.
>
> | Datasets |            | NAGphormer | GOAT    | NodeFormer | SGFormer | SpikeGraphormer | SGHormerVQ |
> |----------|------------|------------|---------|------------|----------|-----------------|------------|
> | CS       | Memory(MB) | 3400       | 12490   | 2822       | **1662**     | 8542            | 1638       |
> |          | Energy(J)        | 0.82       | 1.21    | 0.21       | 0.35     | **0.12**            | 0.16   |
> | Physics  | Memory(MB) | 13628      | 22776   | 7624       | **2944**     | 16414           | 3036       |
> |          | Energy(J)        | 1.86       | 2.35    | 0.46       | 0.78     | **0.27**            | 0.36   |
> | arXiv    | Memory(MB) | 10450      | 21146   | 11988      | **6386**     | 22654           | 7132       |
> |          | Energy(J)        | 1.12       | 9.92    | 0.63       | 0.57     | **0.08**            | 0.18       |
> | Products | Memory(MB) | 7470       | 21974   | -          | **934**      | -               | 13494      |
> |          | Energy(J)        | 16.06       | 143.80    | -          | 8.07     | -               | **3.67**       |

---

> ### Author Response · Authors · 2024-11-22
> **Response to Reviewer NQbN**
>
> > **Q3:** I would also recommend the authors further break down the SVQ module to explore the effects of specific components (e.g., varying quantization levels, spike neuron types, or normalization techniques) could highlight why SVQ is particularly effective and reveal which configurations yield optimal performance.
>
> **R5:** **We have updated more ablation experiments to explore the effects of specific components (spike neuron types and normalization techniques) in section 5.4.** We implement two common spike neurons (IF and LIF), and two normalization algorithms (LayerNorm and STFNorm [6]) to construct 4 baselines. The results are as follows. It shows that PLIF models, which have learnable membrane time constants and synaptic weights, achieve slightly better performance in most cases. The well-designed normalization algorithm for spiking neurons, STFNorm, outperforms the LayerNorm algorithm across all datasets.
>
> | Models                | Pubmed   | CS       | Physics  | arXiv    |
> |-----------------------|----------|----------|----------|----------|
> | SGHormerVQ (IF)        | 81.6     | 92.8     | 96.0     | 71.0     |
> | SGHormerVQ (LIF)       | 79.6     | 92.8     | 96.1     | 72.1     |
> | SGHormerVQ (LayerNorm) | 78.9     | 90.3     | 95.4     | 71.2     |
> | SGHormerVQ (STFNorm)   | **82.6** | 92.3     | **96.5** | **72.4** |
> | SGHormerVQ            | 80.6     | **93.4** | 96.2     | 72.0     |
>
> > **Q4:** I am curious - does the degree of homophily play a role in the SVQ?
>
> **R6:** We discuss SGHormerVQ in the perspective of homophily in Appendix D. The same graph generation strategy is implemented on Cora and Citeseer datasets following the previous study [7]. The following results show the influences of different homophily ratios on predictive results, where the homophily ratio is defined as $\frac{|\{(v,w):(v,w)\in\mathcal{E}\cap y_v =y_w\}|}{|\mathcal{E}|}$. We find that as the homophily ratio decreases, the classification performance initially declines but eventually starts to improve. It implies that SGHormerVQ may achieve strong performances on certain heterophilic graphs.
>
> | **Cora**        |      |      |      |      |      |      |      |      |      |      |      |      |
> |-----------------|------|------|------|------|------|------|------|------|------|------|------|------|
> | Homophily Ratio | 0.81 | 0.74 | 0.68 | 0.63 | 0.59 | 0.52 | 0.46 | 0.42 | 0.38 | 0.32 | 0.28 | 0.25 |
> | ACC             | 84.7 | 76.0 | 72.0 | 66.9 | 64.7 | 59.2 | 64.1 | 64.4 | 66.8 | 68.8 | 72.7 | 76.1 |
>
> | **Citeseer**    |      |      |      |      |      |      |      |      |      |      |      |      |
> |-----------------|------|------|------|------|------|------|------|------|------|------|------|------|
> | Homophily Ratio | 0.74 | 0.65 | 0.58 | 0.53 | 0.48 | 0.41 | 0.36 | 0.32 | 0.28 | 0.24 | 0.20 | 0.18 |
> | ACC             | 74.0 | 65.9 | 56.4 | 54.9 | 53.3 | 48.5 | 51.8 | 53.4 | 54.9 | 61.0 | 70.8 | 80.1 |
>
>
> [1] Fois A, et al. A Spiking Neural Architecture for Vector Quantization and Clustering. ICONIP, 2020.
>
> [2] Liu M, et al. Spiking-diffusion: Vector quantized discrete diffusion model with spiking neural networks. arXiv, 2023.
>
> [3] Chen J, et al. NAGphormer: A tokenized graph transformer for node classification in large graphs. ICLR, 2023.
>
> [4] Wu Q, et al. SGFormer: Single-Layer Graph Transformers with Approximation-Free Linear Complexity. NIPS, 2023.
>
> [5] Yao M et al. Spike-driven transformer v2: Meta spiking neural network architecture inspiring the design of next-generation neuromorphic chips. ICLR, 2024.
>
> [6] Xu M et al. Exploiting spiking dynamics with spatial-temporal feature normalization in graph learning. IJCAI, 2021.
>
> [7] Ma Y et al. Is homophily a necessity for graph neural networks?. ICLR, 2022.

---

> ### Comment · Reviewer_NQbN · 2024-11-24
> **Reply to the Authors**
>
> I would like to thank the authors for their response and have raised my rating. This is a very interesting paper - however I have some more clarifications that would be great if the authors could provide:
>
> 1.It seems from the ablation studies, the use of STFNorm is even better than what the authors are using - can the authors give more insights what they believe is happening and why we are seeing this?
>
> 2. Personally I found your discussion in Appendix A to be very interesting, but too short - would it be possible for the authors to add more details of the comparisons and how they are interpreting the visualizations etc.
>
> 3. I am not very convinced with the empirical results related to the "scalability" of the model as shown by the ogn-products results ( the DNN state-of-the-art is around 90% here). So, I am curious whether this is the right use case for the SGHormerVQ model.

---

> > ### Author Response · Authors · 2024-11-25
> > **Response to Reviewer NQbN**
> >
> > Thank you for the valuable feedback and pointing out the part we can improve further.
> >
> > > **(Q1):** It seems from the ablation studies, the use of STFNorm is even better than what the authors are using - can the authors give more insights what they believe is happening and why we are seeing this?
> >
> > **In our implementation, each module is deliberately designed as simple as possible to highlight the effectiveness of SVQ. Replacing the current normalization layer with advanced normalization variants designed for SNNs would yield better results.** The previous work [1] implements a Spatial-Temporal Feature Normalization as follows:
> > $$
> > \hat{V}[t] = \frac{\rho V_{th}(V[t]-E[V])}{\sqrt{Var[V] + \epsilon}} \\
> > $$
> > $$
> > Y[t] = \lambda\hat{V}[t] + \gamma
> > $$
> > where $V[t]\in\mathbb{R}^{N\times D}$ is denoted as the instant outputting membrane potential of all neurons at the $t$-th step. $\lambda$ and $\gamma$ are two trainable parameters. $\rho$ is a hyper-parameter and $\epsilon$ is a tiny const. The mean and standard deviation are calculated over the feature and temporal dimensions. **We believe that the STFNorm reduces the impact of internal covariate shift while taking into account the threshold voltage in spiking neurons. As described in Eq.(11)-(12), It enables the SVQ to better learn suitable firing patterns based on the input data.**
> >
> > > **(Q2):** Personally I found your discussion in Appendix A to be very interesting, but too short - would it be possible for the authors to add more details of the comparisons and how they are interpreting the visualizations etc.
> >
> > Thank you for the helpful suggestions. **We will update Appendix A with more implementation details of the comparisons, extra visualizations under different configurations and corresponding explanations**. Here, we would like to provide some details about our observations in short. We implement SVQ and its non-spiking counterpart on several datasets in Appendix A. For SVQ (the two rightmost plots in Figure 5 of Appendix A), messages in each propagation step are fed into spiking neurons, whereas the non-spiking counterpart (the leftmost plot in Figure 5) removes these spiking neurons. The visualization results from the non-spiking baseline show that **node embeddings in the same class tend to follow similar distributions.** And the two rightmost plots show **different high-precision embeddings have been converted into the same low-precision spiking vector via spiking neurons, which implies different nodes can be represented by the same codeword.** For $T=4$, increasing the threshold voltage $V_{th}$ from 0.4 to 0.6 leads to lower spike counts/fire rates, which indirectly results in the spiking neurons generating lower-precision spiking vectors. It shows that **different spiking neurons essentially affect the precision of codewords and the number of nodes each codeword can represent.**
> >
> > > **(Q3):** I am not very convinced with the empirical results related to the "scalability" of the model as shown by the ogn-products results ( the DNN state-of-the-art is around 90% here). I am curious whether this is the right use case for the SGHormerVQ model.
> >
> > **R3:** Thank you for these insightful observations. During the experiments, we also found that most of GTs performed significantly worse than the state-of-the-art models on the ogbn-products dataset. It drives us to investigate the underlying causes of this performance gap. **It is consistent with observations reported in a recent study [2] that the existing architecture of GTs derived from graph classification tasks, which simply integrates message passing neural networks with Transformers, may lead to significant performance degradation in node classification tasks.** The performance gap can seen as an inherent issue in existing GTs' architecture. A thorough exploration of the optimal design between MPNN and Transformer modules is beyond the scope of this work. Thank you again for the inspiring advice. It has sparked our curiosity about the better use cases of SGHormerVQ (e.g. graph-level tasks). We plan to explore this question and delve deeper into the SVQ in future work.
> >
> >
> > [1] Xu M, et al. Exploiting spiking dynamics with spatial-temporal feature normalization in graph learning. IJCAI, 2021.
> >
> > [2] Luo Y, et al. Classic GNNs are Strong Baselines: Reassessing GNNs for Node Classification. NIPS, 2024.

---

> > > ### Author Response · Authors · 2024-12-02
> > >
> > > Dear Reviewer NQbN,
> > >
> > > We sincerely appreciate the time and effort you have dedicated to reviewing our response and for raising your score. Your recognition of our work is highly valued and means a great deal to us.
> > >
> > > If you have any additional concerns, please let us know. Thank you again for your time and consideration.
> > >
> > > Best regards,
> > >
> > > Authors

---

### Official Review · Reviewer_V4S6 · 2024-10-30

**Soundness:** 3
**Presentation:** 3
**Contribution:** 2
**Rating:** 5
**Confidence:** 4

**Summary:**

Authors propose SGHormerVQ, a graph transformer model that combines Graph Transformers (GTs) with Spiking Neural Networks (SNNs) through Spiking Vector Quantization (SVQ). SGHormerVQ leverages the spiking mechanism of SNNs to encode compact, low-precision spiking vectors that efficiently capture global graph information. The model was evaluated on multiple node classification datasets, demonstrating superior performance and computational efficiency over various state-of-the-art baselines, with up to a 518x increase in inference speed on some datasets.

**Strengths:**

1.SGHormerVQ introduces SNNs into graph transformers and uses SVQ to address issues of high computational complexity and resource demands in graph data processing. This interdisciplinary model design showcases a novel application of SNNs in graph structures.
2.The paper provides thorough experimental comparisons with various advanced models, and detailed ablation studies support the design decisions behind SGHormerVQ.imited Applicability: The model performs less effectively on datasets with high node degrees (e.g., ogbn-products), suggesting that SVQ may suffer from information loss in highly connected graphs.
Lack of Technical Detail: The practical impact of SVQ and CGSA on computational complexity, hardware requirements, and performance trade-offs is not sufficiently analyzed, potentially limiting the model's real-world applicability.
Unclear Advantage of Innovation: The paper could better articulate the distinct advantages of SVQ over traditional quantization methods, especially regarding its unique application in SNN-based quantization.

**Weaknesses:**

1.limited Applicability: The model performs less effectively on datasets with high node degrees (e.g., ogbn-products), suggesting that SVQ may suffer from information loss in highly connected graphs.
2.Lack of Technical Detail: The practical impact of SVQ and CGSA on computational complexity, hardware requirements, and performance trade-offs is not sufficiently analyzed, potentially limiting the model's real-world applicability.
3.Unclear Advantage of Innovation: The paper could better articulate the distinct advantages of SVQ over traditional quantization methods, especially regarding its unique application in SNN-based quantization.
4.The authors should set up ablation experiments to verify the model improvement on the speed of inference.
5.The description of how the node information of a graph is converted into a codebook is poor.

**Questions:**

Same as Weakness.

---

> ### Author Response · Authors · 2024-11-22
> **Response to Reviewer V4S6**
>
> > **(Q1):** The model performs less effectively on datasets with high node degrees (e.g., ogbn-products), suggesting that SVQ may suffer from information loss in highly connected graphs.
>
> **R1:** In this work, we focus on analyzing the role of spiking neurons in prior spiking graph neural networks from the perspective of vector quantization. To this end, in SVQ, we just adopted the plain spiking neuron, message propagation algorithm, and normalization layer, as shown in Eq.(10)–(12). And experiments demonstrate that SGHormerVQ composed of simple components has achieved significant improvements in both efficiency and predictive accuracy for most cases. Recent studies have proposed more sophisticated alternatives to the above modules. We believe that employing these advanced alternatives with lower quantization errors can effectively address the limitation.
>
> > **(Q2):** The practical impact of SVQ and CGSA on computational complexity, hardware requirements, and performance trade-offs is not sufficiently analyzed, potentially limiting the model's real-world applicability.
>
> **R2:** Thank you for your insightful suggestions. **We update Appendix B to better illustrate the model's real-world applicability.** The energy efficiency analysis is conducted based on three metrics (Memory usage, theoretical energy consumption and inference latency) for showcasing the computational complexity and performance trade-offs of SGHormerVQ. Additionally, we detail the theoretical energy consumption in the formula and discuss the deployment of the model on neuromorphic hardware. The experimental results are as follows:
>
> | Datasets |            | NAGphormer | GOAT    | NodeFormer | SGFormer | SpikeGraphormer | SGHormerVQ |
> |----------|------------|------------|---------|------------|----------|-----------------|------------|
> | CS       | Latency (s) | 0.70        | 5.02    | 0.05       | 0.01     | 0.03            | 0.01       |
> |          | Memory (MB) | 3400       | 12490   | 2822       | 1662     | 8542            | 1638       |
> |          | ACC        | 93.1       | 93.5    | 92.9       | 91.8     | 92.1            | **93.4**   |
> | Physics  | Latency (s) | 1.79       | 10.98   | 0.14       | 0.02     | 0.08            | 0.02       |
> |          | Memory (MB) | 13628      | 22776   | 7624       | 2944     | 16414           | 3036       |
> |          | ACC        | 95.7       | 95.4    | 95.4       | 95.9     | 95.7            | **96.2**   |
> | arXiv    | Latency (s) | 0.78       | 28.27   | 1.17       | 0.10     | 0.30            | 0.08       |
> |          | Memory (MB) | 10450      | 21146   | 11988      | 6386     | 22654           | 7132       |
> |          | ACC        | 70.4       | 72.4    | 59.9       | **72.6**     | 72.0            | 72.0       |
> | Products | Latency (s) | 25.74      | 2416.84 | -          | 24.34    | -               | 20.83      |
> |          | Memory (MB) | 7470       | 21974   | -          | 934      | -               | 13494      |
> |          | ACC        | 73.3       | **82.0**    | -          | 72.6     | -               | 74.8       |

---

> > ### Author Response · Authors · 2024-11-22
> > **Response to Reviewer V4S6**
> >
> > > **(Q3):** The paper could better articulate the distinct advantages of SVQ over traditional quantization methods, especially regarding its unique application in SNN-based quantization.
> >
> > **R3:** Thank you for pointing that out. **Most previous studies in SNN-based quantization can be considered as scalar quantization frameworks.** These methods [1][2][3] focus on decreasing the quantization error/information loss at the element-level. SGHormerVQ takes a step further. By quantizing node representations to a finite set, **SGHormerVQ aims to reduce information loss while ensuring that the quantization results are meaningful at the representation/vector-level.** It achieves that samples in a set are correlated and some set patterns are more likely to occur than others.
> >
> > > **(Q4):** The authors should set up ablation experiments to verify the model improvement on the speed of inference.
> >
> > **R4:** We conducted new ablation experiments using Linformer and Performer as baselines for evaluating the inference latency on 7 datasets. In the implementation, we use the mini-batch partition for training on the ogbn-products dataset. The results are shown below. On small-scale datasets (<20,000), the model's inference speed is slightly slower than that of two classical linear attention modules. As the dataset size increases, the inference latency of SGHormerVQ is comparable to or even better than the others.
> >
> > | Methods               | Cora   | Citeseer | Pubmed | CS     | Physics | arXiv  | Products |
> > |-----------------------|--------|----------|--------|--------|---------|--------|----------|
> > | SGHormerVQ(Linformer) | 0.0034 | 0.0054   | 0.0059 | 0.0073 | 0.0217  | 0.1073 | 19.8036  |
> > | SGHormerVQ(Performer) | 0.0035 | 0.0054   | 0.0073 | 0.0082 | 0.0245  | 1.2137 | 20.9885  |
> > | SGHormerVQ            | 0.0048 | 0.0079   | 0.0095 | 0.0084 | 0.0169  | 0.0866 | 20.8280  |
> >
> >
> > > **(Q5):** The description of how the node information of a graph is converted into a codebook is poor.
> >
> > **R5**: **We will update a more clear description and pseudocode of them to the related section in the final version.** Here, we will briefly outline the process of mapping node information to the codebook. **(i)** For a node, its features are considered as the input current at the $t=0$, $I[0]=R\in\mathbb{R}^D$. **(ii)** $I[t]\in\mathbb{R}^D$ can be obtained by collecting neighborhood messages after $t$ iterations of propagation. **(iii)** $\hat{S}[t]=\Theta(V[t-1], I[t])\in{\\{0,1\\}}^D$ represents the spikes emitted from each output neuron across time. **(iv)** $S=\sum_{t=0}^T{\hat{S}[t]}\in\mathbb{Z}^D$ can be considered as the codeword. Additionally, we provided visualization results of the codebook in Appendix A.
> >
> > [1] Guo Y, et al. Ternary spike: Learning ternary spikes for spiking neural networks. 2024 AAAI.
> >
> > [3] Li J, et al. A graph is worth 1-bit spikes: When graph contrastive learning meets spiking neural networks. ICLR, 2024.
> >
> > [3] Putra R V W, et al. Q-spinn: A framework for quantizing spiking neural networks. IJCNN, 2021.

---

> > > ### Author Response · Authors · 2024-12-02
> > >
> > > Dear reviewer V4S6,
> > >
> > > As the author-reviewer discussion period will close, we would greatly appreciate it if you could review our response and rebuttal, where we updated the additional experiments and more technical details in the final revision to address your concern.
> > >
> > > if you have any additional concerns, please let us know. Thank you for your time and consideration.
> > >
> > > Best regards,
> > >
> > > Authors

---

### Official Review · Reviewer_BudY · 2024-11-02

**Soundness:** 2
**Presentation:** 2
**Contribution:** 2
**Rating:** 5
**Confidence:** 4

**Summary:**

This paper proposes a spike-based graph transformer model SGHormerVQ with spiking vector quantization for graph node classification. It builds on transformer with vector quantization and proposes to leverage spiking vector quantization to replace the pre-defined and fixed codebook. Experiments on various datasets demonstrate the competitive performance and faster inference speed of the proposed method.

**Strengths:**

1. This paper proposes a new spiking vector quantization for efficient graph transformers.

2. The proposed model achieves competitive or superior node classification performance with much faster inference speed than graph transformer baselines.

**Weaknesses:**

1. The motivation for *spiking* vector quantization is not clear enough.

- If it aims at the potential energy efficiency of SNNs, there is no discussion on the issue of deployment on neuromorphic hardware, because SNNs rely on it to obtain real energy efficiency and hardly have efficiency on GPU. Since spiking vector quantization is only a small part of the model and other parts like normalization, MPNN, and self-attention are real-valued artificial neural networks, the model is not a pure SNN model and has problems for deployment. It is also not proper to mark the model as spike-based since some major parts are not spike-based.

- If it just wants to solve the problems of VQ on GPU, why consider spiking neurons? The process is similar to previous random feature propagation, what if directly use it without the spiking function? No experiment demonstrates any advantage of considering the spiking property.

2. The claim “518$\times$ faster inference speed compared to other GTs” is not proper since SGFormer has a similar inference time. It is also unclear why inference speed is largely improved.

3. For the proposed model, spiking vector quantization seems to adapt previous random feature propagation to the spiking function setting for implicit codebook, and codebook guided self-attention also follows previous works. Can the authors distinguish the contributions of this work more clearly?

4. There is no theoretical analysis for the proposed method.

**Questions:**

For SNNs, spiking is only one of the properties. Very early works focused on analyzing their stronger computational power induced by the temporal dimension of spiking time [1], and recent works have introduced this thought to graph AI tasks [2]. This paper only considers rate coding that loses a lot of information of spike trains. Is it possible to introduce these spiking time information to better enhance the codebook coding?

[1] Networks of spiking neurons: the third generation of neural network models. Neural Networks, 1997.

[2] Temporal Spiking Neural Networks with Synaptic Delay for Graph Reasoning. ICML 2024.

---

> ### Author Response · Authors · 2024-11-22
> **Response to Reviewer BudY**
>
> Thank you for your time and thorough comments.
>
> > **(W1), (W3):** "Motivation for spiking vector quantization is not clear enough...why consider spiking neurons? The process is similar to previous random feature propagation, what if we directly use it without the spiking function? Can the authors distinguish the contributions of this work more clearly?
>
> **R1:** In the general responses, we restate our observations that converting neighborhood messages in multiple propagation steps into spiking vectors captures the graph structural information while replacing the original real-value representations space with a narrower spiking representation space. Inspired by these observations, we propose the spiking vector quantization to address issues present in previous VQ-based GNNs:
>
> - **The lack of graph inductive bias.** For the vanilla vector quantization block derived from VQ-VAE [1], its predefined codebook doesn't involve any strong inductive biases. The emerging study [2] finds that structrue-aware tokenizers can be learned by adding the extra edge reconstruction loss. Benefiting from the characteristics of spiking neurons, SVQ actively captures the dynamic during the message propagation to generate a codebook with graph inductive bias.
>
> - **Codebook collapse.** By learning a spiking function rather than pre-defining a codebook, SVQ can be seen as finding the optimal quantizer to map nodes into corresponding codewords. It effectively addresses the problem of codebook collapse.
>
> **In Appendix A, we perform SVQ on Cora, Citeseer and Pubmed datasets, and provide the visualization results of rate coded embeddings.** As depicted in Figure 5 of Appendix A, directly performing random propagation without the spiking function on graph datasets generates high-precision representations for each node. **Feeding the messages generated during multiple propagation steps into spiking neurons will generate the low-precision tokens of nodes, that nodes in the same class tend to be represented by the same spiking vector.**
>
> > **(W1):** "there is no discussion on the issue of deployment on neuromorphic hardware...It is also not proper to mark the model as spike-based since some major parts are not spike-based."
>
> **R2:** **For SGHormerVQ, the spiking vector quantization can be deployed on the neuromorphic hardware.** Notably, the SVQ can be decoupled with other modules, and the codewords corresponding to nodes can be pre-calculated before the inference phase rather than looking up entries in the codebook.  It enables SGHormerVQ to generate and store the codebook with low energy consumption. **Essentially, our model revisits the role of spiking neurons in the perspective of vector quantization and exhibits how spiking vector quantization results can be incorporated into GNNs.** Therefore, we still refer to SGHormerVQ as the spike-driven architecture.

---

> ### Author Response · Authors · 2024-11-22
> **Response to Reviewer BudY**
>
> > **(W2):** The claim "518x faster inference speed compared to other GTs" is not proper since SGFormer has a similar inference time. It is also unclear why inference speed is largely improved.
>
> **R3:** Thank you for your helpful advice. Compared to GOAT, SGHormerVQ with better performance infers faster than this VQ-based GT by 518x on the Physics dataset. We apologize for the confusion and have revised this unclear description. In addition, we provided a comprehensive energy efficiency analysis to evaluate memory usage, inference latency and theoretical consumption of GTs in Appendix B.
>
> **The pre-calculated codebook in SVQ** and **the linear-time Transformer guided by codebook** bring the significant improvement in inference speed.
>
> - In SVQ, the codewords corresponding to nodes can be pre-calculated by feeding features into spiking neurons before the inference phase, rather than looking up entries in the codebook during the inference process.
>
> - As described in the line 299, The complexity of CGSA is $O(NBd)$. The size of the codebook, $B$, also affects the inference speed of SGHormerVQ. As depicted in Figure 4, **the codebook generated by spiking neurons is much smaller in size compared to predefined codebooks.** In some cases, the codebook size $B$ is even smaller than the embedding dimension $d$, which enables the inference latency of our model to be slightly lower than SGFormer $O(Ndd)$.
>
> > **(W3):** Can the authors distinguish the contributions of this work more clearly?
>
> **R4:** We would like to explain the core ideas of this work in detail. For SGHoremrVQ, **one of the main contributions lies in proposing an efficient spike-driven vector quantization (SVQ) method for graph data.** Its core idea involves feeding neighborhood messages from multiple propagation steps into spiking neurons to generate codewords .
>
> It is worth noting that **the message propagation algorithms in SVQ are model-agnostic. Replacing random feature propagation with sophisticated message-passing models (GCN, SAGE and GAT) is feasible.** We believe that combining advanced message-passing mechanisms with spiking neurons can better capture the structural information of graphs and generate codebooks with graph inductive bias. To balance the efficiency and predictive performance of SGHormerVQ, this work adopts a relatively simple implementation.
>
> The architecture design of a Codebook-Guided Transformer is not the primary goal of this work. Instead, **compared to previous VQ-based GTs, our model actively injects the graph inductive bias to generate codebook and indirectly incorporates graph structural information into the Transformer.**
>
> > **(W4):** There is no theoretical analysis for the proposed method.
>
> **R5:** As mentioned in the general response, SGHormerVQ is largely inspired by proposed observations. In SGHormerVQ, we are more focused on revisiting spiking graph neural networks in the perspective of vector quantization rather than solving theoretical problems regarding SGNNs. The theoretical insight mainly comes from previous studies [1][2][3].
>
> > **(Q1)** For SNNs, spiking is only one of the properties. Very early works focused on analyzing their stronger computational power induced by the temporal dimension of spiking time, and recent works have introduced this thought to graph AI tasks. This paper only considers rate coding that loses a lot of information of spike trains. Is it possible to introduce these spiking time information to better enhance the codebook coding?
>
> **R6:** Thank you for the inspired suggestion. It drives us to explore the spiking vector quantization based on temporal coding. **The visualization results of decoding outputs from RGSNN and SVQ are demonstrated in Figure 6 of Appendix C.** Since GRSNN is designed for link prediction tasks and the properties of edges are plain on node classification datasets, we also assign random features to nodes except for the embedding of relations. The outputs combined edge embeddings containing the temporal delay information with node embeddings will be fed into a mean aggregator to generate the predictive results. The results show that GRSNN integrating temporal delays in output spikes does provide more expressive spiking vectors. **However, reconstructing more expressive node embeddings via spiking time as additional dimension conflicts with SVQ, which aims at mapping different nodes into the same low-precision spiking vectors.**
>
> [1] Li J, et al. A graph is worth 1-bit spikes: When graph contrastive learning meets spiking neural networks. ICLR, 2024.
>
> [2] Eshraghian J K, et al. Training spiking neural networks using lessons from deep learning. arXiv, 2021.
>
> [3] Gerstner W, et al. Neuronal dynamics: From single neurons to networks and models of cognition. 2014

---

> > ### Comment · Reviewer_BudY · 2024-11-26
> >
> > I thank the authors for their responses. Yet, many of my questions are not answered clearly. Regarding the motivation, it is still unclear what is the advantage of the spiking function. For the mentioned two problems, "The lack of graph inductive bias" and "Codebook collapse", message propagation and learnable codebook can deal with the problems and it is unclear why spiking is necessary or what is the advantage of the claimed "low-precision tokens" (no experiment demonstrates how it influences performance). For deployment on neuromorphic hardware, only the SVQ part can be deployed so the energy efficiency would be very limited (no analysis is presented as well) and it will have a large cost for data transfer between different hardware, so the energy efficiency can not be a valid argument for the method. Since the main contribution lies in the introduction of the spiking neuron, there are critical issues that are not solved yet. So, I keep my rating currently.

---

> > > ### Author Response · Authors · 2024-11-29
> > > **Response to Reviewer BudY**
> > >
> > > Thank you for the feedback.
> > >
> > > > **(Q):** it is still unclear what is the advantage of the spiking function. For the mentioned two problems, "The lack of graph inductive bias" and "Codebook collapse", message propagation and learnable codebook can deal with the problems and it is unclear why spiking is necessary or what is the advantage of the claimed "low-precision tokens" (no experiment demonstrates how it influences performance).
> > >
> > > We would like to elaborate on the role of spiking neurons in SGHormerVQ. **Most VQ-based methods with the learnable codebook do not deal with the problem of codebook collapse. It should be emphasized that the codebook generation scheme driven by spiking neurons, referred to as SVQ in our work, mitigates the problem of codebook collapse.** This phenomenon has been reported in previous works [1][2], and the results in Table 3 also demonstrate that previous VQ-based GNNs also suffer from this issue. To better illustrate how spiking neurons bridge the message propagation with the codebook, **we provide additional visualization results in Figure 5 of Appendix A.** As depicted in Figure 5, the full-precision messages generated in each propagation step (the three leftmost plots in Figure 5) can be transformed into spiking embeddings via spiking neurons. And then the rate-coded vectors of the nodes (the rightmost plot in Figure 5) are calculated by summing the above spiking outputs over $T$ propagation step. **Compared with the full-precision embeddings, rate-coded vectors (low-precision tokens) not only capture the dynamic of the message propagation, but also naturally represent nodes from a finite set of integer vectors which are considered as codewords in our model. The spiking neuron/spiking function is the foundation for achieving the above operation.**
> > >
> > > **The inference of SVQ on predictive performances has been provided in Table 4 of Section 5.4.** We compare SGHormerVQ with the baseline with a fixed codebook after removing SVQ modules on Pubmed, CS, Physics and ogbn-arxiv datasets. The results are as follows. SGHormerVQ outperforms the baseline across four datasets. It suggests that encoding the node embeddings from multiple propagation steps into rate-coded vectors via spiking neurons leads to a better codebook containing the structural knowledge of each node.
> > >
> > > | Models              | Pubmed       | CS           | Physics      | ogbn-arxiv   |
> > > |---------------------|--------------|--------------|--------------|--------------|
> > > | SGHormerVQ (w/o SVQ) | 79.8±0.4     | 93.2±0.3     | 95.0±0.4     | 70.7±0.7     |
> > > | SGHormerVQ          | **80.6±0.5** | **93.4±0.1** | **96.2±0.0** | **72.0±0.1** |

---

> > > > ### Author Response · Authors · 2024-11-29
> > > > **Response to Reviewer BudY**
> > > >
> > > > > **(Q):** For deployment on neuromorphic hardware, only the SVQ part can be deployed so the energy efficiency would be very limited (no analysis is presented as well) and it will have a large cost for data transfer between different hardware, so the energy efficiency can not be a valid argument for the method.
> > > >
> > > > Thank you for raising the discussion about energy efficiency. **We have provided a comprehensive analysis of theoretical energy consumption, maximum memory usage and inference latency among GTs in Table 5 of Appendix B.** The results are as follows. For GTs, the majority of the energy consumption overhead comes from the Transformer module in GTs. Different from those SNNs [3] in the field of computer vision, whose performances are comparable with the full-precision Transformer-based architecture, simply introducing spiking neurons into graph transformers often leads to significant performance degradation [4].
> > > >
> > > > We believe that the role of spiking neurons in message passing networks has been insufficiently explored. **In this work, we show that spiking neurons not only can be solely considered as the low-power units of a model, but also effectively encode the dynamics during message propagation into the codebook to reduce the computational complexity of the Transformer.** The results show that SGHormerVQ with better performances achieves the fastest inference speed with acceptable energy consumption. We believe SGHormerVQ provides a new perspective to advance spike-driven energy-saving GTs on graph data.
> > > >
> > > > | Datasets |            | NAGphormer | GOAT    | NodeFormer | SGFormer | SpikeGraphormer | SGHormerVQ |
> > > > |----------|------------|------------|---------|------------|----------|-----------------|------------|
> > > > | CS       | Latency(s) | 0.7        | 5.02    | 0.05       | **0.01**     | 0.03            | **0.01**       |
> > > > |          | Memory(MB) | 3400       | 12490   | 2822       | 1662     | 8542            | **1638**       |
> > > > |          | Energy(J)  | 0.82       | 1.21    | 0.21       | 0.35     | **0.12**            | 0.16  |
> > > > | Physics  | Latency(s) | 1.79       | 10.98   | 0.14       | **0.02**     | 0.08            | **0.02**       |
> > > > |          | Memory(MB) | 13628      | 22776   | 7624       | **2944**     | 16414           | 3036       |
> > > > |          | Energy(J)  | 1.86       | 2.35    | 0.46       | 0.78     | **0.27**            | 0.36   |
> > > > | arXiv    | Latency(s) | 0.78       | 28.27   | 1.17       | 0.10     | 0.30            | **0.08**       |
> > > > |          | Memory(MB) | 10450      | 21146   | 11988      | **6386**     | 22654           | 7132       |
> > > > |          | Energy(J)  | 1.12       | 9.92    | 0.63       | 0.57     | **0.08**            | 0.18       |
> > > > | Products | Latency(s) | 25.74      | 2416.84 | -          | 24.34    | -               | **20.83**      |
> > > > |          | Memory(MB) | 7470       | 21974   | -          | **934**      | -               | 13494      |
> > > > |          | Energy(J)  | 16.06       | 143.80    | -          | 8.07     | -               | **3.67**        |
> > > >
> > > > [1] Mentzer F, et al. Finite scalar quantization: Vq-vae made simple. ICLR, 2024.
> > > >
> > > > [2] Fifty C, et al. Restructuring Vector Quantization with the Rotation Trick. NIPS, 2024.
> > > >
> > > > [3] Yao M et al. Spike-driven transformer v2: Meta spiking neural network architecture inspiring the design of next-generation neuromorphic chips. ICLR, 2024.
> > > >
> > > > [4] Sun Y, et al.  SpikeGraphormer: A High-Performance Graph Transformer with Spiking Graph Attention. arXiv, 2024.

---

> > > > > ### Author Response · Authors · 2024-12-02
> > > > >
> > > > > Dear Reviewer BudY,
> > > > >
> > > > > We genuinely appreciate your comments and feedback, and your insights help us further enhance our work. We believe the additional experiments and visualization results we provided highlight the necessity of spiking neurons in our work.
> > > > >
> > > > > As the author-reviewer discussion period will close, if you have any additional concerns, please let us know. Thank you for your time and consideration.
> > > > >
> > > > > Best Regards,
> > > > >
> > > > > Authors

---

### Official Review · Reviewer_oc9C · 2024-11-04

**Soundness:** 1
**Presentation:** 2
**Contribution:** 2
**Rating:** 3
**Confidence:** 4

**Summary:**

In this work, the authors proposed SGHormerVQ, which is a graph transformer with spiking neural networks. SGHormersVQ utilizes spiking vector quantization with a codebook. According to the authors’ experiments, they could achieve 518x faster inference speed compared to other models.

**Strengths:**

- Through experiments and competitive results
- Incorporating graph Transformer and spiking neural networks through spiking vector quantization

**Weaknesses:**

This study presents a method to combine spiking neural networks and graph Transformers through spiking vector quantization to reduce the computational complexity, but it seems that the vector quantization technique using codebook has serious concerns in terms of soundness.

This study utilizes a method to reduce the computational complexity of self-attention through vector quantization using a codebook. Vector quantization is implemented using spiking neurons as quantization functions. This causes the codebook to change dynamically, which is expressed in the manuscript as the term “implicit codebook.” However, this is actually just a spike train, which is the activation value of a spiking neuron, and it is difficult to apply a quantization technique that uses a codebook because it changes dynamically.

In addition, in general, in order to improve efficiency due to quantization, the representation space of the codebook should be narrower than the representation space of the activation. However, in this study, since these two are used identically, it is correct to say that there is no improvement due to the codebook. In this paper, vector quantization using an implicit codebook simply processes information with spike activation. It is unreasonable to view the improvement as being due to a codebook vector quantization. The main reason for the improvement is believed to be due to the characteristics of spiking neural networks that process information with binary spikes. The authors need to clarify this.

Furthermore, a significant portion of the manuscript is devoted to explaining existing research. It is necessary to concisely revise it and devote more parts to explaining the proposed model. Overall, it is recommended that the authors revise the manuscript.

- Miscellaneous
  - Redundant definition of abbreviations - SNNs, GT, SVQ
  - Used before abbreviations are defined - GNNs (line 107)
  - Used undefined abbreviations - EMA (line 148)

**Questions:**

- Why is the title SGHormerVQ?
- What algorithm was used for learning?
- What surrogate function was used?
- What is the effect of the norm function in Equation 11?
- What is R in g{V,e,X} → g{V,e,R} in Figure 2?
- What does U in Equation 13 represent? It is a one-hot matrix, so what information does it contain?
- Line 305: “we construct random features” → What does it mean?
- Is SVQ the only part that uses spiking neurons? It seems that information is not processed by spikes in other parts. Then, how can this model work on neuromorphic hardware?

---

> ### Author Response · Authors · 2024-11-22
> **Response to Reviewer oc9C**
>
> We appreciate the reviewer for detailed comments and suggestive feedback. We want to clarify some misunderstandings that caused some of your concerns.
>
> > **(W1):** "However, this is actually just a spike train, which is the activation value of a spiking neuron, and it is difficult to apply a quantization technique that uses a codebook because it changes dynamically."
>
> **R1:** First of all, we present the definition of rate coded outputs. $\hat{S}[t]=\Theta(V[t-1], I[t])\in{\\{0,1\\}}^D$ is a time-varying vector that represents the spikes emitted from each output neuron across time, where $\Theta(\cdot)$ is the Heaviside function. Let $S=\sum_{t=0}^T{\hat{S}[t]}\in\mathbb{Z}^D$ be the spike count from each output neuron, which can be obtained by summing $\hat{S}[t]$ over $T$ time steps. For $D=1, T=3$ and the input sequence is $I=[0.6, 0.5, 1.0]$, the rate coded output will be $S=2$. As described in Eq.(10)-(12), SVQ collects the messages $M[t]$ during the multiple propagation steps as input currents $I[t]$ across time, and converts the input sequences to integers. Considering spiking neurons as quantizers, we can utilize low-precision rate coded outputs (we refer to them as "low-precision spiking vectors" in the paper) to represent different nodes. This is also one of the contributions of our paper. We add additional visualization results in Appendix A to detail the role of spiking neurons during the feature propagation process.
>
> > **(W1)** " the representation space of the codebook should be narrower than the representation space of the activation. However, in this study, since these two are used identically"
>
> **R2:** **In SGHormerVQ, the representation space of the implicit codebook is much narrower than the representation space of the activation.** For $D=3, T=2$, and node features can be seen as the input at $t=0$, $I[0]\in\mathbb{R}^3$, SVQ generates the input sequence $I\in\mathbb{R}^{2\times3}$ and mapped it into a implicit codebook $\tilde{C}=\\{(0,0,0),(0,0,1),(0,0,2),...,(2,2,2)\\}$, where $|\tilde{C}|=(T+1)^D=27$. SVQ quantizes node representations to a finite set of codewords. More details have been explained in lines 255-257.
>
> > **(W1)** "vector quantization using an implicit codebook simply processes information with spike activation. It is unreasonable to view the improvement as being due to a codebook vector quantization. The main reason for the improvement is believed to be due to the characteristics of spiking neural networks that process information with binary spikes. The authors need to clarify this."
>
> **R3:** **We believe that the improvement in predictive performances of SGHormerVQ is brought from the fusion of SVQ and CGSA modules.**  We will clarify the role of each module in SGHormerVQ:
> - **SVQ:** Figure 1 shows that SVQ mapps different node representations into same low-precision spiking vectors by combining the random feature propagation with spiking neurons. Those nodes represented by same spiking vector can be regarded as having similar neighborhood structures. **These lower-space vectors require less storage space, which brings an improvement in efficiency.**
> - **CGSA:** By replacing the original key matrix with spiking vectors, the vanilla attention from nodes to nodes can be transformed into a linear-time attention from nodes to grouped node sets. **The CGSA module effectively captures long-range structural information and generate more expressive node embeddings**.

---

> ### Author Response · Authors · 2024-11-22
> **Response to Reviewer oc9C**
>
> > **(Q1):** Why is the title SGHormerVQ?
>
> **R4:** For differentiating the existing GT method [1] and highlighting the spiking vector quantization module, we named our method SGHormerVQ (Spiking GrapH transfORMER via a Vector Quantization variant).
>
> > **(Q2), (Q3):** What algorithm was used for learning? What surrogate function was used?
>
> **R5:** The backpropagation algorithm is used to update the model parameters. Besides, as mentioned in the line 186, we adopt surrogate gradients during error backpropagation to address the issue of zeros gradients caused by non-differentiable functions in Eq.(12). In the implementation, we adopt the Sigmoid function as the surrogate function.
>
> > **(Q4): What is the effect of the norm function in Equation 11?**
>
> **R6:** As mentioned in the line 269, the $Norm(\cdot)$ in Equation 11 is responsible for normalizing output messages to the range of threshold membrane potential $V_{th}$.
>
> > **(Q5), (Q7):** What is $R$ in g{V,e,X} → g{V,e,R} in Figure 2? Line 305: "we construct random features" → What does it mean?
>
> **R7:** In line 254, we detail that $R$ is denoted as the random features, which is a $D$-dimension matrix sampled from the uniform distribution. And random features are widely used to improve the expressiveness of MPNNs [2][3].
>
> > **(Q6):** What does U in Equation 13 represent? It is a one-hot matrix, so what information does it contain?
>
> **R8:** $U$ can be seen as the indices for where the codewords in the rate coded embeddings $S$ ended up in the unique matrix $C$. And $C$ can be seen as removing duplicate vectors from $S$. Specifically, as describe in Eq.(13), the rate coded embeddings $S$ can be decomposed into the codebook $C\in\mathbb{Z}^{B\times D}$ and the one-hot matrix $U\in{\\{0,1\\}}^{N\times B}$, where $B$ is the size of reconstruted codebok, $B\ll|\tilde{C}|$ and $D$ is the dimension of random features.
>
> > **(Q8):** Is SVQ the only part that uses spiking neurons? It seems that information is not processed by spikes in other parts. Then, how can this model work on neuromorphic hardware?
>
> **R9:** In SGHormerVQ, the spiking vector quantization module driven by spiking neurons can be deployed on specific neuromorphic hardware. We notice that different from some existing spiking-driven Transformer methods in computer vision [4], directly converting the output embeddings from Transformer blocks or GNN blocks into the spiking format will seriously impair the predictive performance of models. In SGHormerVQ, the reconstructed codebook $C$ is derived from the rate coded embeddings, which means we can feed random features into neuromorphic hardware to generate the codewords corresponding to nodes with lower energy consumption. Additionally, we provided a comprehensive energy efficiency analysis in Appendix B.
>
> [1] Wu Q, et al. SGFormer: Single-Layer Graph Transformers with Approximation-Free Linear Complexity. NIPS, 2024.
>
> [2] Yao M, et al. Spike-driven transformer. NIPS, 2024.
>
> [3] Puny O, et al. Global attention improves graph networks generalization. arXiv, 2020.
>
> [4] Dasoulas G, et al. Coloring graph neural networks for node disambiguation. arXiv, 2019.

---

> > ### Comment · Reviewer_oc9C · 2024-11-28
> >
> > I appreciate the author's response. However, my concerns are not resolved sufficiently. The method of spiking neurons as a quantizer is not well explained and is hard to accept. The implicit codebook is simply the same as the activations of spiking neurons. It is more reasonable to consider this as binarization rather than a codebook for quantization. I think a graph transformer model with spiking vector quantization, which is the main approach in this work, is not valid. Therefore, I would keep my initial score.

---

> > > ### Author Response · Authors · 2024-11-29
> > > **Response to Reviewer oc9C**
> > >
> > > Thank you for the feedback.
> > >
> > > > **(Q):** The method of spiking neurons as a quantizer is not well explained and is hard to accept. The implicit codebook is simply the same as the activations of spiking neurons. It is more reasonable to consider this as binarization rather than a codebook for quantization. I think a graph transformer model with spiking vector quantization, which is the main approach in this work, is not valid.
> > >
> > > We would like to clarify some misunderstandings. **Codewords in the implicit codebook are not the activations of spiking neurons.** For a node, the calculation of its codeword can be summarized as follows: (i) generating the neighborhood message embeddings $m\in\mathbb{R}^D$ from $T$ propagation steps, $\\{m_0, m_1,...,m_T\\}$. (ii) converting these high-precision embeddings into spiking embeddings $\hat{s}\in\\{0,1\\}^D$ through a spiking neuron $SN(\cdot)$, $\\{\hat{s_0},\hat{s_1},...,\hat{s_T}\\}=\\{SN(m_0), SN(m_1),...,SN(m_T)\\}$. (iii) calculating the summation of its spiking embeddings over $T$ propagation steps as the codeword, $s=\sum_{t=0}^T{\hat{s}_t}, s\in\mathbb{Z}^D$. This is a widely used rate coding mechanism in SNNs, known as the spike count. **The codewords are integer vectors rather than binary vectors.** In the revision, **we refer to codewords as the term "rate-coded vectors" to distinguish them from the spiking activations which consist of 0 and 1 elements, and we update more explanation in Section 4.1.**
> > >
> > > In addition, **we visualize the process from floating-point messages in multiple propagation steps to integer rate-coded vectors in Figure 5 of Appendix A.** The visualization results show that nodes can be represented by a finite set of integer vectors by capturing the multi-step message propagation patterns and converting them into spike counts via SVQ. Empirical experiments in Table 1 and Table 5 show that replacing the original full-precision node representations with these rate-coded vectors in CGSA does reduce the computational complexity while verifying the effectiveness of SGHormerVQ.

---

> > > > ### Author Response · Authors · 2024-12-02
> > > >
> > > > Dear Reviewer oc9C,
> > > >
> > > > As the author-reviewer discussion period will close, we would greatly appreciate it if you could review our further response and rebuttal, where we addressed your concern regarding more explanations about the codebook generation.
> > > >
> > > > We earnestly hope that you will consider updating your rating to reflect the current state of the discussion. If you have any additional concerns, please let us know. Thank you for your time and consideration.
> > > >
> > > > Best Regards,
> > > >
> > > > Authors

---

> > > > > ### Comment · Reviewer_oc9C · 2024-12-02
> > > > >
> > > > > Thanks for the detailed answer.
> > > > >
> > > > > It doesn't really matter whether the codebook is binary or the accumulated spike count.
> > > > > My question is what does the implicit codebook mean in the proposed quantization?
> > > > > The proposed method seems to be that the floating point input is simply input to the spiking neuron, and it is accumulated.
> > > > > Then, the accumulated spike counts are called the implicit codebook.
> > > > > This method works the same even without the concept of a codebook, and this is a method unrelated to the quantization method using a codebook.
> > > > > Could you please explain this further?
> > > > > What is the difference with and without a codebook?

---

> > > > > > ### Author Response · Authors · 2024-12-02
> > > > > > **Response to Reviewer oc9C**
> > > > > >
> > > > > > > **(Q):** My question is what does the implicit codebook mean in the proposed quantization? ... the accumulated spike counts are called the implicit codebook. This method works the same even without the concept of a codebook, and this is a method unrelated to the quantization method using a codebook. Could you please explain this further? What is the difference with and without a codebook?
> > > > > >
> > > > > > Thank you for your further feedback. **The accumulated spike count is not equivalent to the implicit codebook.** Here, we will elaborate following concepts in our paper: **accumulated spike counts (rate-coded vectors)** $S$, **the implicit codebook** $\hat{C}$, **the reconstructed codebook** $C$. For $N=3, T=2, D=2$, where $T$ is the number of time steps and $D$ is the dimensionality of embeddings, the accumulated spike counts embeddings of nodes, $S\in\mathbb{Z}^{N \times D}$, are as follows:
> > > > > > $$
> > > > > > S=
> > > > > > \begin{bmatrix}
> > > > > > 0,\quad 1 \\\\
> > > > > > 1,\quad 2 \\\\
> > > > > > 0,\quad 1 \\\\
> > > > > > \end{bmatrix}
> > > > > > $$
> > > > > > **The implicit codebook is the latent space of the spike count embedding.** For $T=2, D=2$, the implicit codebook can represented as follows:
> > > > > > $$
> > > > > > \hat{C} = \\{[0, 0], [0, 1], [0, 2], [1, 0], [1, 1], [1, 2], [2, 0], [2, 1], [2, 2]\\}
> > > > > > $$
> > > > > > For the spike count embeddings, $\forall s\in\hat{C}$, and the size of this finite vector set (implicit codebook) is $|\hat{C}|=(T+1)^D=3^2=9$. This is because, for $D=2$, each dimension can take three possible spike counts (0, 1, 2) across $T=2$ time steps. At last, the accumulated spike count embeddings can be decomposed as follows:
> > > > > > $$
> > > > > > S=
> > > > > > \begin{bmatrix}
> > > > > > 0,\quad 1 \\\\
> > > > > > 1,\quad 2 \\\\
> > > > > > 0,\quad 1 \\\\
> > > > > > \end{bmatrix}
> > > > > > =UC=
> > > > > > \begin{bmatrix}
> > > > > > 1,\quad 0 \\\\
> > > > > > 0,\quad 1 \\\\
> > > > > > 1,\quad 0 \\\\
> > > > > > \end{bmatrix}
> > > > > > \begin{bmatrix}
> > > > > > 0,\quad 1 \\\\
> > > > > > 1,\quad 2 \\\\
> > > > > > \end{bmatrix}
> > > > > > $$
> > > > > >
> > > > > > where $U\in\\{0, 1\\}^{N\times B}$ is a one-hot matrix, $C\in\mathbb{Z}^{B\times D}$ is the reconstructed codebook matrix, $B$ is the size of the reconstructed codebook, and $B=2<|\hat{C}|=9$. In a nutshell, the implicit codebook denotes the latent space of spike count embedding, which is determined by $T$ and $D$.
> > > > > >
> > > > > > As described above, we do not explicitly create a codebook $\hat{C}$ to represent all codewords from the discrete latent space or replace the embeddings with codewords by measuring the distances. **Converting floating point embeddings into spike count embeddings and decomposing the outputs automatically generate the subset $C$ of the implicit codebook $\hat{C}$. This avoids the issue of explicitly defining the codebook that contains a large number of unused codewords, which is one of the core contributions of the paper.**
> > > > > >
> > > > > > Furthermore, in Eq.(17)-(19) of Section 4.2, we replace the Key matrix in the Transformer with the node embeddings, which can be decomposed into a product of the codebook and the one-hot matrices, to effectively reduce the cost to linear time. **It shows that revisiting the codebook/vector quantization from the perspective of spikes provides an effective way to represent nodes from narrower latent spiking embeddings space.**

---

### Author Response · Authors · 2024-11-22
**General response**

Dear Area Chairs and Reviewers,

We thank the reviewers for their thorough reviews and constructive suggestions. We have carefully considered your comments and incorporated more discussion into the manuscripts. Besides, we add  new experiment results to further strenghthen our contributions. Here is a summary of the key revisions:


1. We have corrected some typo errors in the revision.
2. We provided more explainations about our observations and experimental results to detail the motivation and key novelty for our method. The updated contents are highlighted in orange.
3. For better addressing the raise problems, we conduct additional experiments and update results in the Appendix A~D.

Below we restate our main motivations and contributions to facilitate the discussions. In this study, we investigate the role of spiking neurons in current SGNNs and present the following three observations:
- **[O1]:** For message propagation-based models, the trained embeddings of nodes within the same class tend to exhibit similar distributions.
- **[O2]:** Messages received from the multiple propagation steps can be converted into the same spike trains via spike neurons. Decoding the outputs into firing rates or spike counts, different nodes can be represented by the same low-precision vectors.
- **[O3]:** The precision of decoding node representations can be adjusted by spiking neurons.

Corresponding visualization results are demonstrated in Appendix A. The above observations drive us to push the field of spiking graph neural networks, our contributions include:
- **[C1] New Vector Quantization Variant:** Constructing spiking-based vector quantization mitigates codebook collapse and injects graph inductive bias.
- **[C2] New Linear-time Graph Transformer:** Utilizing message propagation patterns guides Graph Transformer in generating node representations with linear complexity.
- **[C3] Performance:** SGHormerVQ is able to achieve performance on par with both advanced baselines while demonstrating significant efficiency advantages in terms of codebook usage, inference latency, memory usage, and energy consumption.

In the following individual response, we provide answers to each raised weakness/question point and will incorporate the suggestions and new results in our revised paper.

---

> ### Author Response · Authors · 2024-11-29
>
> Dear Area Chairs and Reviewers,
>
> We sincerely thank all the reviewers for their valuable feedback. In response to the remaining concerns, we have revised the paper to better highlight its significance. The new changes are marked in **red**, while the previous changes are marked in **orange**. The specific modifications in this final revision are as follows:
>
> 1. We provide extra visualization results and corresponding explanations in Appendix A. Besides, we update these visualization results to better distinguish floating point node embeddings and integer rate-coded vectors.
> > (NQbN) "Personally I found your discussion in Appendix A to be very interesting, but too short - would it be possible for the authors to add more details of the comparisons and how they are interpreting the visualizations etc."
>
> 2. We compare the high-precision embeddings during the message propagation and the corresponding rate-coded vectors in Figure 5 of Appendix A to highlight the necessity of spiking neurons.
> > (BudY) "it is unclear why spiking is necessary or what is the advantage of the claimed "low-precision tokens"
>
> 3. We refer to the spike count outputs obtained by summing spiking embeddings over $T$ propagation steps as **rate-coded vectors**, to explicitly differentiate them from spiking activations which consist of 0 and 1 elements.
> > (oc9C) "The implicit codebook is simply the same as the activations of spiking neurons. It is more reasonable to consider this as binarization rather than a codebook for quantization."

---

### Author Response · Authors · 2024-11-25
**Gentle Reminder to Reviewers**

Dear Reviewers,

We sincerely appreciate your insightful review and feedback comments. All comments have greatly enhanced our paper.

**As the author-reviewer discussion deadline is approaching, we would like to check if you have any other remaining concerns about our paper.** We have faithfully responded to all your comments and revised our paper according to your suggestions. If you have any further questions, we will be happy to answer them. And we eagerly await your feedback on whether our responses have satisfactorily resolved your concerns. We understand that you have a demanding schedule, and we appreciate the time and effort you dedicate to reviewing our paper.

Kind regards,

Authors

---

### Author Response · Authors · 2024-12-04
**Rebuttal Summary**

Dear AC and reviewers,

First and foremost, we greatly appreciate all reviewers who voluntarily spend their time in order to improve the quality of scientific research in our field. The insightful comments and constructive suggestions are indeed helpful for improving our work. During the rebuttal period, we have tried to address **ALL** the reviewers’ concerns adequately and believe most of the concerns have been covered. To facilitate grasping the key points of the entire rebuttal, we provide the following summary:

- **More explanations regarding observations of existing spiking graph neural networks (SGNNs).** We update a number of visualization results in Appendix A of the revision to help readers better understand the necessity of spiking neurons in SGNNs and the motivation of SGHormerVQ.
- **More implementation details of SVQ.** We restate how SVQ reconstructs the integer codebook from floating point node embeddings step by step. These implementation details are provided in the related sections of the revision to clarify potential misunderstandings.
- **Comprehensive analysis of energy efficiency.** For those concerns about the energy consumption, memory usage and inference latency raised by most reviewers, we have provided a detailed energy efficiency analysis comparing the state-of-the-art GTs with SGHormerVQ in Appendix B.
- **Additional technical details.** We conduct additional experiments including the effects of different components, the choice of coding mechanisms, and performances on heterophilic datasets, to demonstrate the practical applicability of SGHormerVQ. Corresponding results are presented in Section 5.4 and Appendix C-D.

We confidently believe that this work dives deep into the field of SGNNs, our contributions include:

- A series of observations analyze the role of spiking neurons in existing SGNNs.

- The spiking vector quantization module (SVQ) represents nodes from narrower latent spiking embeddings space to alleviate the codebook collapse.

- The Graph Transformer module (CGSA) with linear complexity generates node embeddings indirectly guided by message propagation patterns.

- Empirical experiments show that SGHormerVQ effectively addresses the issues presented in previous VQ-based GNNs. SGHormerVQ achieves comparable or even faster inference speed compared to other GT baselines.

We understand that the discussion during the rebuttal period is limited due to time constraints. **We hope our response can spark more discussion during the AC-reviewer phase, to confirm whether our rebuttal has adequately addressed reviewers' concerns.**

Best regards,

the Authors

---

### Meta-Review · Area_Chair_ZtLy · 2024-12-16

**Metareview:**

This work applies spiking neural networks to improve the efficiency of graph transformers. The empirical success in reducing computational overhead and the presentation quality are appreciated by all reviewers.

The discussion has highlighted a few key issues which remain unresolved. The concept of implicit codebook and other key novelties should be explained in more detail in the writing. How the spiking vector quantization mechanism helps to mitigate codebook collapse and graph inductive bias should be better clarified. Overall, the submission has merit to be published and can benefit from another round of revision to address these issues.

**Additional Comments On Reviewer Discussion:**

3 of the 4 reviewers participated in the discussion. The other reviewer was communicated by email but not responsive.

The empirical strength has been further verified during the rebuttal, and the authors have provided more evidence of energy consumption etc. and updated their submission.

The reviewers still have shared concerns on the novelty (combining existing mechanisms), and how the spiking neural network quantization addresses codebook collapse and graph inductive bias.

The authors claimed the implicit codebook as a core contribution but there is less details, than there should be, to explain the concept and basic properties, and to distinguish it with existing quantization mechanisms.

---

### Decision · Program_Chairs · 2025-01-22

Reject